# Expanding the landscape of aging via orbitrap astral mass spectrometry and tandem mass tag integration

Gregory R. Keele [1,5], Yue Dou[2,5], Seth P. Kodikara[2], Erin D. Jeffery [2], Dina L. Bai[2], Erik Hultenius[3], Zichen Gao[3], Joao A. Paulo [4], Steven P. Gygi [4], Xiao Tian[3] & Tian Zhang [2] ✉

Aging results in a progressive decline in physiological function due to the deterioration of essential biological processes. While proteomics offers insights into aging mechanisms, prior studies are limited in proteome coverage and lifespan range. To address this, we integrate the Orbitrap Astral Mass Spectrometer with the multiplex tandem mass tag (TMT) technology to profile the proteomes of cortex, hippocampus, striatum and kidney in the C57BL/6JN mice, quantifying 8,954 to 9,376 proteins per tissue (12,749 total). Samples spanned both sexes and three age groups (3, 12, and 20 months), representing early to late adulthood. To improve TMT quantitation accuracy, we develop a peptide-spectrum match-based filtering strategy that leverages resolution and signal-to-noise thresholds. Our analysis uncovers distinct tissue-specific patterns of protein abundance, with age and sex differences in the kidney and primarily age-related changes in brain tissues. We also identify both linear and non-linear proteomic trajectories with age, revealing complex protein dynamics over the adult lifespan. Integrating our findings with early developmental proteomic data from brain tissues highlights further divergent age-related trajectories, particularly in synaptic proteins. This study provides a robust data analysis workflow for Orbitrap Astral–based TMT analysis and expands the proteomic understanding of aging across tissues, ages, and sexes.

Progressive deterioration in fundamental biological processes results in aging-related decline, leading to increased risk of disease and mortality[1]. Loss of physiological integrity is the primary risk factor for major human pathologies, including cancer and neurodegenerative diseases[1–4]. Transcriptional correlates with aging-related decline have been well-studied, through both bulk[5] and single-cell[6] RNA-seq experiments. Though there have been some studies of aging in murine models[7–11], age-related changes at the protein level are less understood, and display poor correspondence with transcript-level aging changes

across a range of murine tissues[12–14]. A recent discovery-based proteomic study in 10 tissues from C57BL/6J mice revealed age differences between adult (8 months) and late midlife ages (18 months)[15]. Surprisingly, brain tissues had relatively fewer age and sex differences than other tissues. However, this study provided a limited map of age differences with only two age groups and approximately only 5000 proteins quantified in each tissue. We hypothesized that the lack of detected age differences in brain tissues was due in part to insufficient proteome coverage and insufficient variability in surveyed age groups.

[1]GenOmics, Bioinformatics, and Translational Research Center, RTI International, Research Triangle Park, NC, USA. [2]Department of Biochemistry and Molecular Genetics, School of Medicine, University of Virginia, Charlottesville, VA, USA. [3]Sanford Burnham Prebys Medical Discovery Institute, La Jolla, CA, USA. [4]Department of Cell Biology, Harvard Medical School, Boston, MA, USA. [5]These authors contributed equally: Gregory R. Keele, Yue Dou. ✉e-mail: cgm8ck@virginia.edu

The dynamic range of the proteome is vast, with the cellular proteome spanning seven orders of magnitude[16]. Fractionation reduces sample complexity and thereby enables greater peptide and protein coverage in proteomic analysis[17,18]. However, the identification and quantitation of many low-abundance peptides still suffer from the limited sensitivity of mass spectrometers. In the Orbitrap Astral mass spectrometer, the Asymmetric Track Lossless (Astral) analyzer enables up to 200 Hz acquisition of high-resolution accurate mass (HRAM) MS/MS spectra. The parallel acquisition using the Orbitrap and Astral analyzers permits full scans with a high dynamic range and resolution and fast and sensitive HRAM MS/MS scans, expanding proteome coverage and ensuring quantitative accuracy[19–22]. Recent advances in tandem mass tag (TMT)-based proteomic technology have enabled multiplexed sample preparation and data acquisition, allowing complex experimental designs with no loss in quantitative integrity[23,24]. The high sensitivity of the Astral analyzer also enables the separation of TMTpro reporter ions and the utilization of TMTpro 18-plex for sample multiplexing. Moreover, limited previous studies have shown the use of TMT on the Orbitrap Astral Mass Spectrometer[25,26].

Here, we integrate the Orbitrap Astral Mass Spectrometer with TMT technology to enable a multiplexed proteomic analysis with increased proteome coverage. To improve the age dimension resolution, we profile the proteomes of three brain tissues (cortex, hippocampus and striatum) from male and female mice aged 3, 12, and 20 months, representing approximately young adulthood to early late life (~20–60 years in humans). In addition to the brain tissues, the kidney with its extensively characterized sex differences[12,15] is included. To enhance quantitative accuracy, we develop a rigorous peptide-spectrum match-based filtering strategy using resolution and signal-to-noise thresholds and quantify ~9000 proteins per tissue. Our analysis reveals distinct tissue-specific patterns of protein abundance. The dataset highlights both age and sex differences in the kidney, while revealing age differences but limited sex differences in brain tissues. Notably, we identify both linear and non-linear (i.e., continuous and non-continuous) proteomic changes with age, revealing complex protein dynamics over the lifespan. We integrate our dataset with early developmental proteomic data from brain and kidney tissues, and further highlight divergent age-related trajectories, particularly in brain synaptic proteins. This study broadens the proteomic landscape of aging and highlights the exceptional capability of the Orbitrap Astral Mass Spectrometer to capture age-related molecular changes with remarkable depth.

## Results

For each tissue, three replicates of male and female C57BL/6JN mice at the ages of 3, 12, and 20 months were multiplexed in a TMTpro 18-plex (Supplementary Data 1). The samples were fractionated into 96 fractions using offline HPLC and recombined into 24 samples[17]. Twelve out of 24 samples were then analyzed using the Orbitrap Astral Mass Spectrometer (Fig. 1a).

### Enhancing TMT quantitation accuracy with Orbitrapol astral mass spectrometry through peptide filtering

From these data, we noticed that the peptides with high intensities can yield erroneous results if their ion intensities reach $10^6$, leading to distorted and non-ideal detector response. Reporter ion intensities from a single scan for the peptide GTGASGSF from Histone 1.5 are shown as an example (Fig. 1b–d). Instead of 18 reporter ions, 12 peaks were detected in the reporter ion m/z range, with twelve of the carbon and nitrogen isotope versions of the TMTs coalesced into six peaks (Fig. 1b, Supplementary Fig. 1a). As an example, the peak merged identified with m/z 130.1386 is 0.0038 Da to the right of 130 N and 0.0025 Da to the left of 131 C (Fig. 1c). Using 0.001 Da peak match tolerance (PTM) window for reporter ion quantitation resulted in the quantitation of 3 reporter ions (dark green) compared to 12 reporter

ions using 0.003 Da PTM window. However, this adjustment failed to mitigate the impact of the inaccurate peptide quantitation on the protein quantitation (Fig. 1d). This erroneous quantitation of this peptide led to a distortion of the quantitation of Histone 1.5. Removing this peptide from protein quantitation revealed an age effect on protein H1.5 (Fig. 1e). Detector saturation caused by high-intensity ions (i.e., ion saturation) has been widely reported in time-of-flight (TOF) mass spectrometers and has also been observed in other mass spectrometry platforms, including triple quadrupoles and trap-based instruments[27]. In this study, we hypothesized that the low resolution resulting from ion saturation contributed to the inaccurate quantification of TMT reporter ions.

To systematically investigate the saturation effects in these datasets, we developed a C# program to extract the resolution of each TMT reporter ion in each scan. The program recorded the resolution, intensity, and noise of each reporter ion with a peak match tolerance of 0.001 Da for each MS2 spectrum from the raw file, enabling detailed downstream analysis. Subsequently, the resolutions of the 18 TMT reporter ions associated with peptides used for protein quantitation were analyzed (https://doi.org/10.6084/m9.figshare.28016501). A resolution of 45 K is required to distinguish the carbon and nitrogen isotopologs of the TMT tags, which have a mass difference of 0.006 Da[24,28]. To assess the impact of resolution, the $\log_{10}$ ratio of the total signal-to-noise (S/N) across 18 channels for each MS2 spectrum was plotted for the four tissue types, focusing on cases where any of the 18 reporter ions fell below the 45 K resolution threshold (Fig. 2a). The results revealed a bimodal distribution in resolution corresponding to low and high S/N groups. Less than full resolution (i.e., poor resolution) in the low-S/N group was attributed to low-intensity ions, whereas poor resolution in the high-S/N group was attributed to ion saturation. We investigated further whether detector saturation was an issue in TMT datasets collected using an Orbitrap analyzer. We downloaded raw files from PXD018886[29] and PXD032843[30], checked the MS2 and MS3 data collected in an Orbitrap analyzer, and detected no PSMs with saturated effects, indicating that this is a specific issue to the Orbitrap Astral analyzer. We next tested whether reducing the sample input would alleviate this problem by injecting approximately half the amount used in the original analysis of hippocampus tissue. This resulted in a reduced number of quantified peptides and fewer peptides with S/N greater than 1440 across 18 channels (Supplementary Fig. 1b). The number of PSMs with any of the resolutions of TMT reporter ions below 45 K decreased from 10,572 to 2963, still impacting 1383 proteins. Additionally, the number of quantified proteins dropped from 9370 to 8762 (Supplementary Fig. 1c). We also tested lower injection amounts, however, the PSMs with low resolution could not be fully eliminated (Supplementary Fig. 1d–f).

To assess how resolution and signal-to-noise (S/N) could influence the detection of quantitative differences due to biological factors of interest, we tested for differences between age groups and sex using regression (Methods) in peptides and proteins (summed from their component peptides). Peptides with low resolutions and low intensities were associated with significant differences (i.e., effects) (Supplementary Fig. 2a, b). Comparing the regression coefficient for a protein to the coefficients for its peptides revealed a low correlation for low S/N peptides, which had coefficients closer to zero and thus would likely bias protein abundance towards the null of no detectable differences (Supplementary Fig. 2c–f). The measurement of low-intensity peptides is more likely to be compromised by noise and interference. Excluding these peptides with low resolutions and low intensities improves the accuracy of protein quantitation and facilitates the detection of biologically relevant quantitative differences[31].

To avoid bias from low-S/N and low-resolution peptides, we estimated abundance levels for each protein from its component peptides after filtering out peptides that did not have full resolution and S/N lower than 1440 for 18 samples. The protein abundance estimate for an

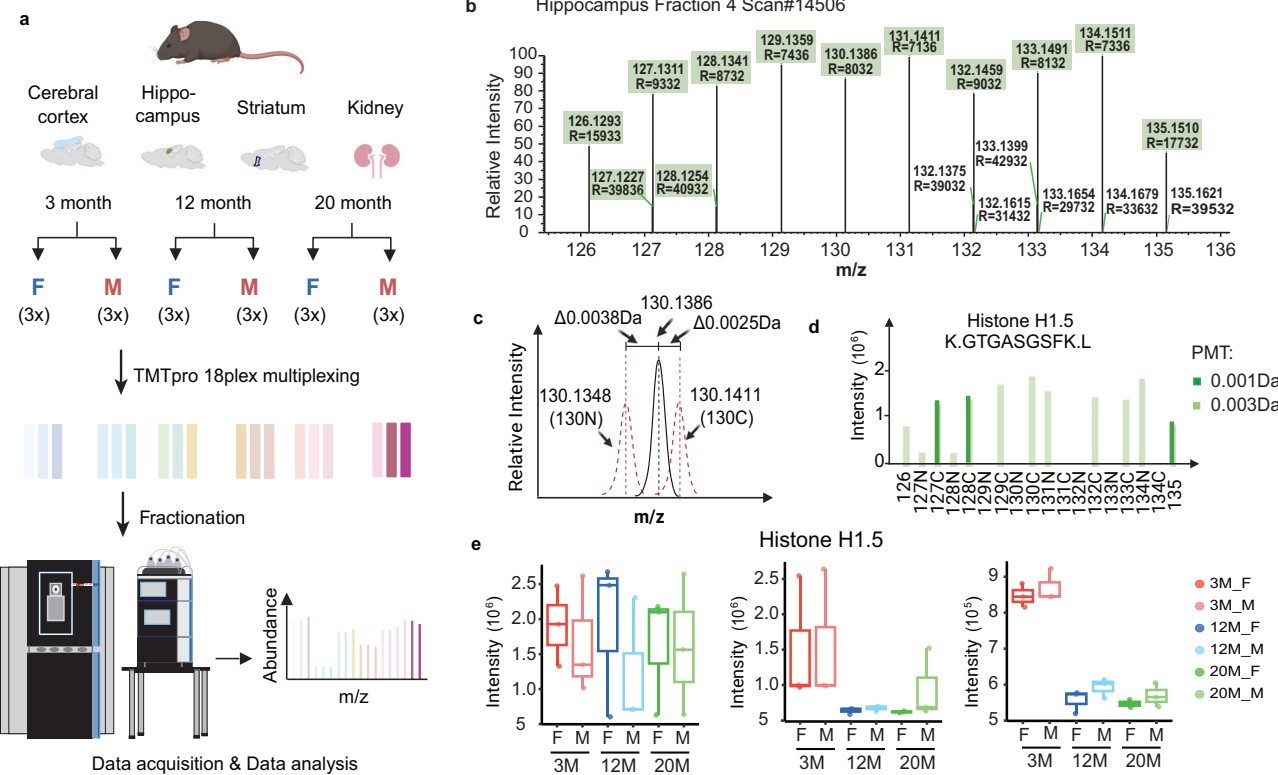

**Fig. 1 | Peptide filtering enhances TMT quantification accuracy with Orbitrap astral mass spectrometry. a** Overview of the workflow. For each tissue, three male and female C57BL/6JN mice at ages 3, 12, and 20 months were multiplexed. Samples were digested, labeled using TMTpro 18plex, combined, fractionated and recombined into 24 fractions, of which 12 were analyzed on an Orbitrap Astral Mass Spectrometer. **b** Reporter ion region of a peptide, GTGASGSF, from Histone 1.5, in the hippocampus tissue is shown. TMT reporter ions identified using a Da peak match tolerance (PMT) of 0.003 Da were highlighted in green boxes. **c** Ion saturation leads to inaccurate measurement of TMT reporter ions. The peak at m/z 130.1368 shifted by 0.0038 Da from 130 N and 0.0025 Da from 131 C due to ion saturation, showing coalescence within a narrow range. **d** Quantitative impact of saturated ions on peptide using different peak match tolerance (PTM) windows. A PMT window of 0.001 Da for quantitation enabled detection of 3 reporter ions (dark green) compared to 12 reporter ions (light green) detected with a 0.003 Da window, though it did not correct for the inaccurate quantitation of this peptide. **e** Excluding this peptide from the analysis corrected for the distortion in Histone 1.5 protein quantitation, revealing a significant age effect. The left panel shows protein quantitation with the peptide using PMT 0.003 Da, while PMT 0.001 Da was used for the middle panel. The right panel shows protein quantitation after removing the peptide with ion saturation effects; PMT 0.001 Da was used. Box-and-whisker plots show the intensities of Histone 1.5 protein. Boxes indicate the interquartile range (25–75th percentile) with the median marked by a horizontal line; whiskers extend to the most extreme points within 1.5× the interquartile range. Individual data points represent biological replicates. N = 3 biological replicates per group.

individual was then the sum of their S/N levels for the high confidence peptides. ~6% of all the PSMs were removed by resolution filtering, while ~20% of all the PSMs were removed due to being of low intensity. The resolution and intensity filtering led to ~2.5% and 9%–10% drop in the number of unique peptides, respectively (Fig. 2b, c). The filtering resulted in an ~5% reduction in protein numbers (Fig. 2c). The filtered peptides are from ~80% of all the quantified proteins (Fig. 2d), and the filtering impacts their quantitation. More than 1/3 of all the quantified proteins were impacted by the resolution filtering, with saturated ions impacting greater than 20% of all quantified proteins (Fig. 2d).

After filtering, our dataset possessed a higher level of coverage at both the peptide and protein levels compared to TMT datasets collected on other mass spectrometers[15,17,29], with quantitation of 9260 proteins in cortex, 9376 in hippocampus, 8954 in striatum, and 9058 in kidney (Supplementary Data 2, Supplementary Fig. 3a). Across the four tissues, 12749 proteins were quantified. The most common overlap/unique category in the UpSet plot was 5778 proteins observed in all four tissues, followed by 1588 in just kidney and 1541 in all three brain tissues (Supplementary Fig. 3b). These numbers are higher (by ~1000 or more per tissue) than recent aging protein studies in mice[10,11,15], which included both data-independent acquisition (DIA) and other (TMT experiments.

## Tissue, age, and sex are major drivers of protein abundance variation

We next sought to characterize variation in protein abundance across tissues, age groups, and sexes. Examining correlation at the level of tissue type, we observed strong correlation among the brain tissues (Supplementary Fig. 4a). The brain tissues were less correlated with kidney tissue, which displayed notable sex-based variation (Supplementary Fig. 4a).

We then performed principal component analysis (PCA) to determine whether age and sex drive the observed variation (Methods). The first principal component (PC1) primarily distinguished the brain tissues from kidney tissue (Supplementary Fig. 4b). PC2 primarily distinguished the striatum from the other tissues and PC3 distinguished the hippocampus and cortex from each other and the other tissues (Supplementary Fig. 4b, c). Cumulatively, tissue type explained 97.5% of the variation (Supplementary Fig. 4d). PC4 (0.4%) aligned cleanly with sex, which was strongest in the kidney as expected[12,15] (Supplementary Fig. 4e). Age correlated with PCs 5 and 6, representing 0.5% of the overall variation (Supplementary Fig. 4f). These results support the potential for tissue-specific signals as well as the large-scale presence of sex and age differences at the level of individual proteins.

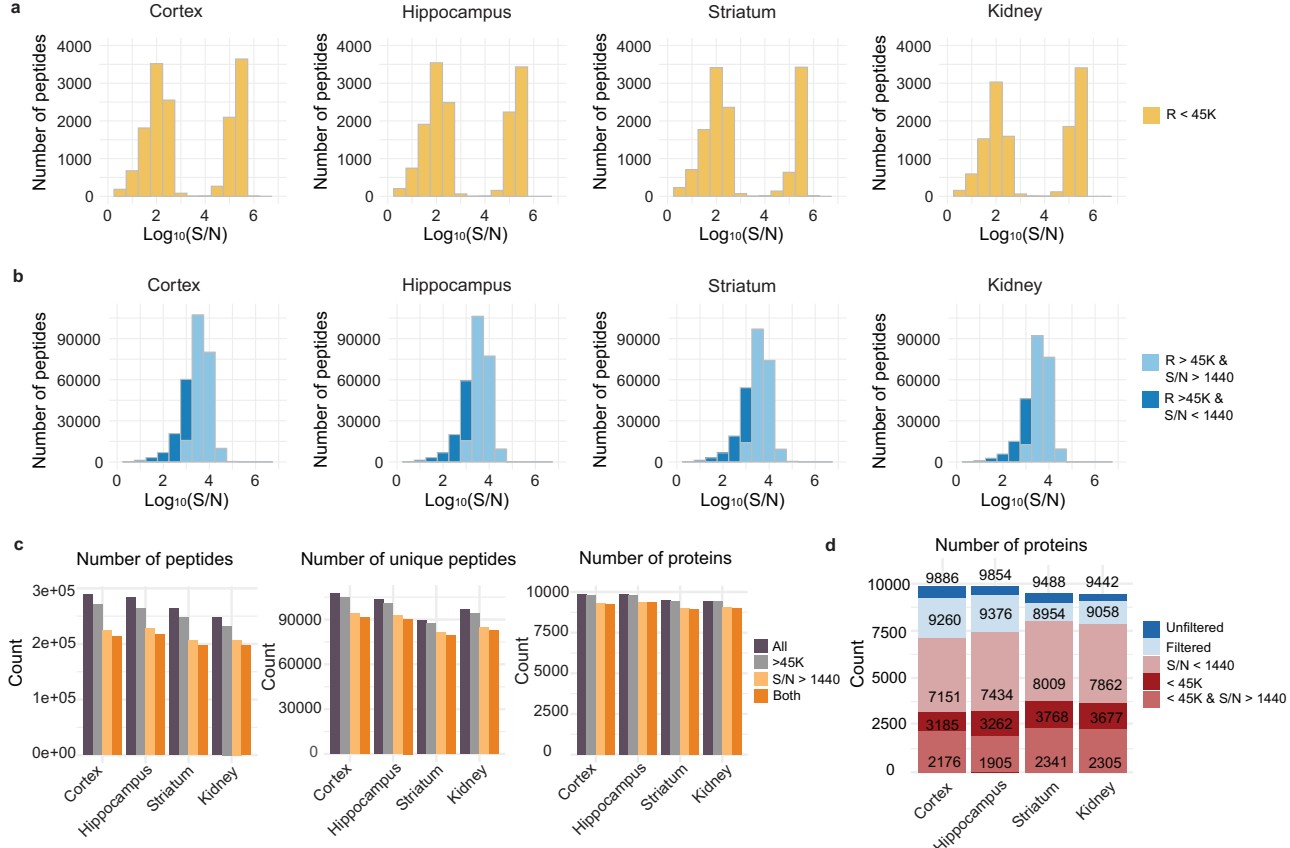

**Fig. 2 | Impact of resolution and Signal-to-Noise (S/N) Filtering on peptide and protein quantitation. a** The $\log_{10}$ ratio of the total S/N across 18 channels for each MS2 spectrum with any resolution below the 45 K threshold was plotted for four tissues, showing a bi-modal distribution for low and high S/N. **b** The intensity distribution of each MS2 spectrum after resolution-based filtering. **c** Resolution and S/N filtering effects on peptide and protein quantitation. Number of peptides, unique peptides and proteins were plotted for each tissue respectively. R > 45 K, S/N > 1440, or both were used as filters. **d** Impact of filtering on protein quantitation. A total of 9886, 9854, 9488, and 9442 proteins were quantified in the cortex,

hippocampus, striatum, and kidney tissues, respectively. After applying all filtering criteria, the number of quantified proteins was reduced to 9260, 9376, 8954, and 9058 in the respective tissues. Filtering based on an S/N threshold of 1400 removed peptides from 7151, 7434, 8009, and 7862 proteins. Filtering using a resolution threshold of 45 K removed peptides from 3185, 3262, 3768, and 3677 proteins. Additionally, applying a resolution cutoff of 45 K removed peptides from 2176, 1905, 2341, and 2305 proteins in the cortex, hippocampus, striatum, and kidney tissues, respectively.

## Protein age differences are highly prevalent in brain and kidney tissues

To identify proteins with age and sex effects, we tested for individual proteins with abundance differences between age groups, sexes, and age-by-sex groups within a single tissue (Methods, Supplementary Data 3). We first modeled age as both a continuous term, thus estimating a single coefficient and assuming a continuous (i.e., linear) trend with age.

The number of proteins with continuous age differences ranged from 1109 (in hippocampus) to 4196 (in striatum) (Fig. 3a, Supplementary Fig. 3c). In the brain, sex differences were relatively minor, ranging from 41 (in cortex) to 242 (in striatum) (Fig. 3b, Supplementary Fig. 3c). Kidney had 2242 proteins with age differences and notably 4272 with sex differences. Examples include male-specific expression of the Y chromosome-encoded EIF2S3Y (Supplementary Fig. 3d), a positive control, and increased expression of the glutathione enzyme GSTA4 in females (Supplementary Fig. 3e), consistent with previous observation[32]. Gene set enrichment analysis[33] (GSEA) was performed using gene sets defined by continuous age differences (increasing and decreasing), age trends categories, and sex differences (increased in males or females) are provided in Supplementary Data 4-7, which include results based on gene ontologies and KEGG pathways. GSEA findings in kidney included increased abundance with age for actin

cytoskeleton proteins, which is highly consistent with previously observed aging-related actin remodeling in the kidney's podocytes[12], and immune-related proteins (Supplementary Fig. 5). We confirm other previous patterns such as increased ribosomal proteins in males and large-scale metabolic differences in kidneys between the sexes[15]. GSEA results for brain tissues revealed a general pattern of increases in metabolic proteins (e.g., GSTA4) and decrease in synaptic proteins. Cortex was an outlier based on having over 3000 categorical age-by-sex differences (Supplementary Fig. 3c), exemplified by USP24 (Supplementary Fig. 3f), a ubiquitin specific peptidase involved in autophagy and potentially relevant to Parkinson disease susceptibility in humans[34]. Overall, these proteins were enriched in functions related to endoplasmic reticulum, synapses, and neurons more broadly (Supplementary Data 4).

## Consistency in protein aging changes across studies

We compared age and sex coefficients from our study to those reported by Keele et al.[15], which also employed a TMT design but utilized a Lumos mass spectrometer. C57BL/6 J mice, instead of C57BL/6JN, from only two age groups were profiled: 8 and 18 months-old. Another distinction was that cerebellum was profiled rather than cortex. We note that there are fixed genetic differences distinguishing the C57BL/6 J and C57BL/6JN sub-strains, including 34 SNPs and 2

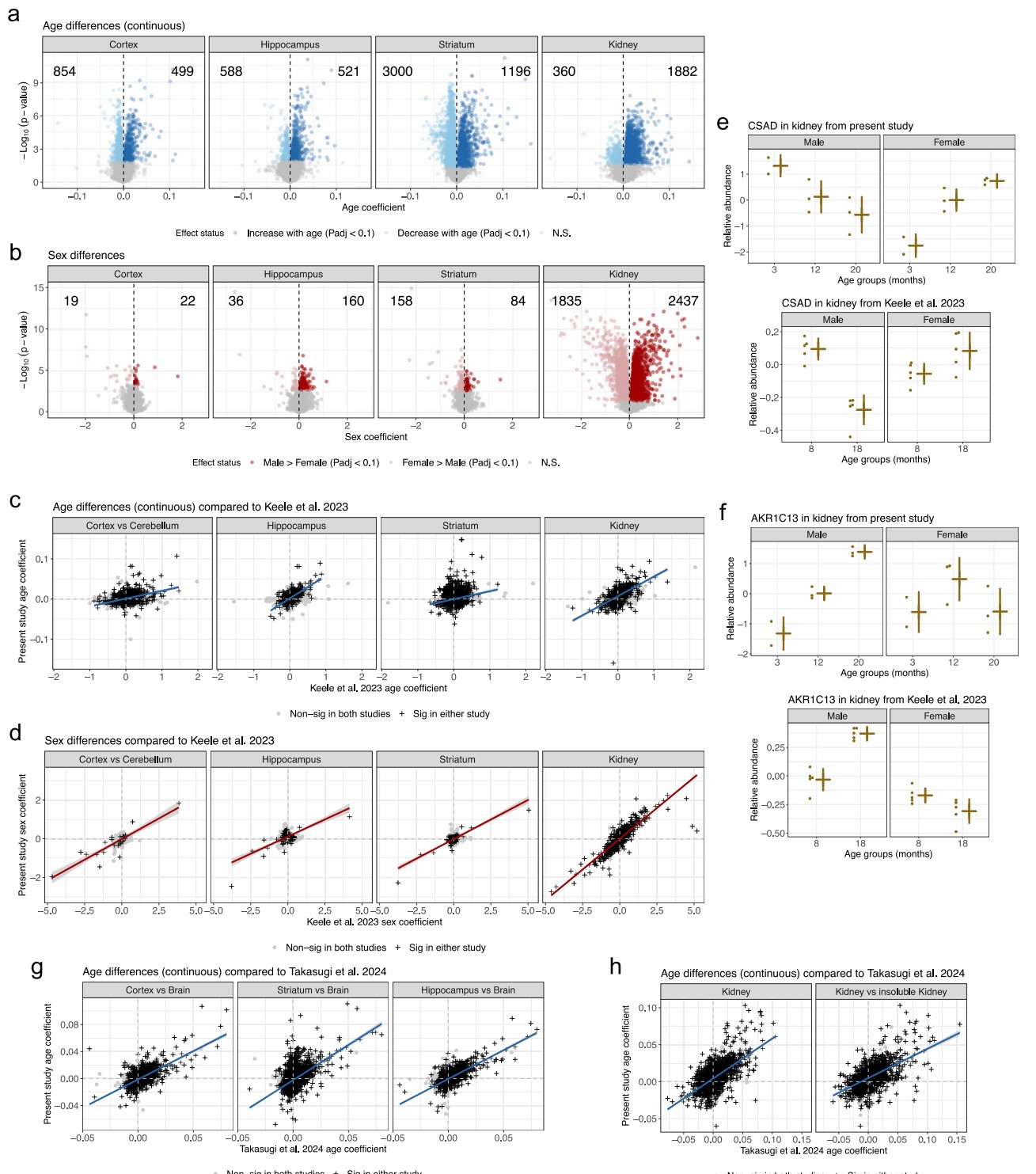

indels in coding regions[35]. Differing phenotypes have been observed, including those related to behavioral changes with age[36]. Comparisons between genetic backgrounds may contribute to discordance in age and sex differences between studies, though notably, consistent signals allow us to detect robust changes across C57BL/6 sub-strains.

Keele et al.[15] only detected 10 proteins in hippocampus, 0 protein in striatum and 496 proteins in kidney tissue with age differences. To test whether the greater number of proteins with age differences in our study were systematically driven by the increased detection of low-abundance proteins, we compared the S/N of proteins unique to our study with those quantified in both studies. We found that most

proteins detected with age differences in our study were quantified in both studies. Furthermore, the proteins observed in only this study had lower average S/N and a lower rate of age differences detected compared to proteins observed in both studies (Supplementary Fig. 6a–c). These findings suggest that the identification of more proteins with age differences in our study is likely a result of both improved statistical power due to observing a greater dynamic range of ages (three age groups rather than two) and improved proteome coverage. Despite the difference in the number of proteins with detected age differences, we observed consistent age coefficients for proteins with detected differences in either study (Fig. 3c). Similarly,

**Fig. 3 | Age and sex differences detected in present study and comparisons to other adult aging proteomics studies. a** Volcano plots of continuous age differences for four tissues. Age differences in individual proteins tested with F tests, followed by FDR adjustment. Counts of statistically significant differences (FDR < 0.1) included for both directions of coefficient. **b** Volcano plots of sex differences for four tissues. Sex differences in individual proteins tested with F tests, followed by FDR adjustment. Counts of statistically significant differences (FDR < 0.1) included for both directions of coefficient. **c** Comparison of continuous age coefficients between present study and Keele et al.[15]. Best fit line from regressing age coefficients from present study on those from Keele et al.[15] that were significant in either study (FDR < 0.1) ± SE included for reference. Age differences in individual proteins tested with F tests, followed by FDR adjustment in each study. **d** Comparison of sex coefficients between present study and Keele et al.[15]. Best fit line from regressing sex coefficients from present study on those from Keele et al.[15] that were significant in either study (FDR < 0.1) ± SE included for reference. Sex differences in individual proteins tested with F tests, followed by FDR adjustment in each study. **e** CSAD had a downward trend with age in male kidneys and an upward trend with age in female kidneys, representing a consistent age-by-sex difference in

the present study (top) and Keele et al.[15] (bottom). For the present study, N = 2 biological replicates for both 3-month groups and N = 3 for 12- and 20-month groups. For Keele et al.[15], N = 5 biological replicates for all age-by-sex groups. Mean ± SD indicated for each age-by-sex group. **f** AKR1C13 had a downward trend with age in male kidneys and an up-down trend with age in female kidneys. representing a consistent age-by-sex difference in the present study (top) and Keele et al.[15] (bottom). For the present study, N = 2 biological replicates for both 3-month groups and N = 3 for 12- and 20-month groups. For Keele et al.[15], N = 5 biological replicates for all age-by-sex groups. Mean ± SD indicated for each age-by-sex group. **g** Comparison of continuous age coefficients between present study and Takasugi et al.[14]. Best fit line from regressing age coefficients from present study on those from Takasugi et al.[14] that were significant in either study (FDR < 0.1) ± SE included for reference. Age differences in individual proteins tested with F tests, followed by FDR adjustment in each study. **h** Comparison of sex coefficients between present study and Takasugi et al.[14]. Best fit line from regressing sex coefficients from present study on those from Takasugi et al.[14] that were significant in either study (FDR < 0.1) ± SE included for reference. Age differences in individual proteins tested with F tests, followed by FDR adjustment in each study.

sex differences were also highly consistent in the kidney, as well as for the few that were detected in the brain tissues in either study (Fig. 3d).

We then leverage these two studies to validate complex age-by-sex differences as demonstrated for proteins such as CSAD (Fig. 3e) and AKR1C13 (Fig. 3f) in the kidney. CSAD exhibited a consistently increasing trend in females but a decreasing trend in males across the age groups. In contrast, AKR1C13 showed a steady increase in males, while females displayed a non-monotonic up-down pattern, which only becomes evident with the broader age range. The inclusion of three distinct age groups in our study provided the resolution to capture non-linear age dynamics that would be impossible to detect with only two age groups. The observation of consistent interaction effects across independent studies strongly supports the biological validity of these findings.

We next compared age coefficients to another study, Takasugi et al.[14], which also employed a TMT design with a Lumos mass spectrometer and C57BL/6 J mice, like Keele et al.[15]. Only males were studied, with four animals per four age groups: 6, 15, 24, and 30 months-old, thus including truly geriatric animals (30 months-old) compared to previous studies. Eight tissues were profiled, including brain, as a whole tissue, and kidney, both as whole tissue and low-solubility enriched fractions. Based on our re-analysis of the brain and kidney data from Takasugi et al.[14], we detected far more age differences (1064 in brain, 3225 in kidney, and 1364 in insoluble kidney) than Keele et al.[14], representing similar numbers to the present study (less in the brain tissues and slightly more in kidney) and providing support for our inference that observing more age groups is integral to detecting age changes. As with Keele et al.[15], the Astral mass spectrometer allowed us to detect more lower abundance proteins (Supplementary Fig. 6d–f). Overall, we saw even stronger correspondence between age differences with Takasugi et al.[14] (Fig. 3g, h). These comparisons demonstrate the quality of the data from these studies and their utility as resources for defining proteins that exhibit age-related changes across study and C57BL/6 sub-strains.

### Non-continuous age differences in proteins

Age dynamics may not follow a constant proportional change with age[37,38], and non-linear or non-continuous patterns could be missed using only a linear regression model with a continuous age term. To investigate proteins exhibiting non-continuous age changes and evaluate whether the data were more consistent with a continuous or non-continuous age difference, we expanded our approach to model categorical (*i.e.*, non-continuous) age trends using Bayesian Information Criterion (BIC) to compare non-nested models (Methods). Briefly, a lower BIC represents a better model fit; we categorized age differences as continuous or non-continuous based on the model with a

lower BIC. Overall, we observed more continuous age differences than non-continuous (Fig. 4a). Each brain tissue had more than twice as many continuous age differences (continuous to non-continuous ratios ranging from 2.24 to 3.56), whereas the kidney had a lower rate of continuous age differences comparatively (continuous to non-continuous ratio of 1.24). We then compared to the data from Takasugi et al.[14], which included four age groups and thus potentially possessed greater power to detect non-continuous age trends. Overall trends were highly similar between the studies with the majority of age differences being more consistent with a continuous model and kidney displaying the greatest level of non-continuous age changes (Fig. 4b). Continuous vs non-continuous age trend for proteins with age differences (FDR < 10%) in at least one of the studies was found to be consistent between studies (hypergeometric p-value = 9.24e-9).

We further categorized age trends based on stepwise changes between adjacent timepoints by testing the specific timepoint effect in the categorical age model. For example, a trend of "Up-up" represents a protein that has increased abundance at 12 months relative to 3 months and increased abundance for 20 months relative to 12 months. In all tissues, the most prevalent trend categories were "Down-Down" and "Up-Up", which is consistent with the higher prevalence of continuous age differences. Examples of proteins with different stepwise age trends include LPXN's increasing continuous trend in kidney, DAD1's decreasing continuous trend in striatum, Rras2's decreasing but non-continuous trend in striatum, and an immunoglobulin variable chain's "Up-Down" trend in kidney (Fig. 4c–f). We performed GSEA for gene sets defined by categorical age trends (Supplementary Data 4–7). Generally, the strongest enrichment signal was found for "Down-Down" and "Up-Up" trends, consistent with the patterns observed with continuous age (synapse-related proteins and metabolism, respectively). Some of the age trends that are less consistent with a continuous age pattern potentially reveal important aging dynamics, such as Rac protein signaling and specific immune ontologies (viral response and T cell mediated) in striatum proteins with "Flat-Up" trends (Supplementary Data 6). These represent systematic age changes that appear to occur primarily between mid-to-late adult life.

### Protein age difference patterns across brain tissues

To investigate tissue-specific and non-tissue-specific age and sex effects, we performed hierarchical clustering using 5778 proteins detected in all four tissues. This analysis revealed that the brain tissues clustered together based on both sex and age differences, while the kidney tissue exhibited a distinct pattern (Fig. 5a, b). Building on this, we focused on data from the brain tissues to identify age-related differences that were either consistent across tissues or unique to specific

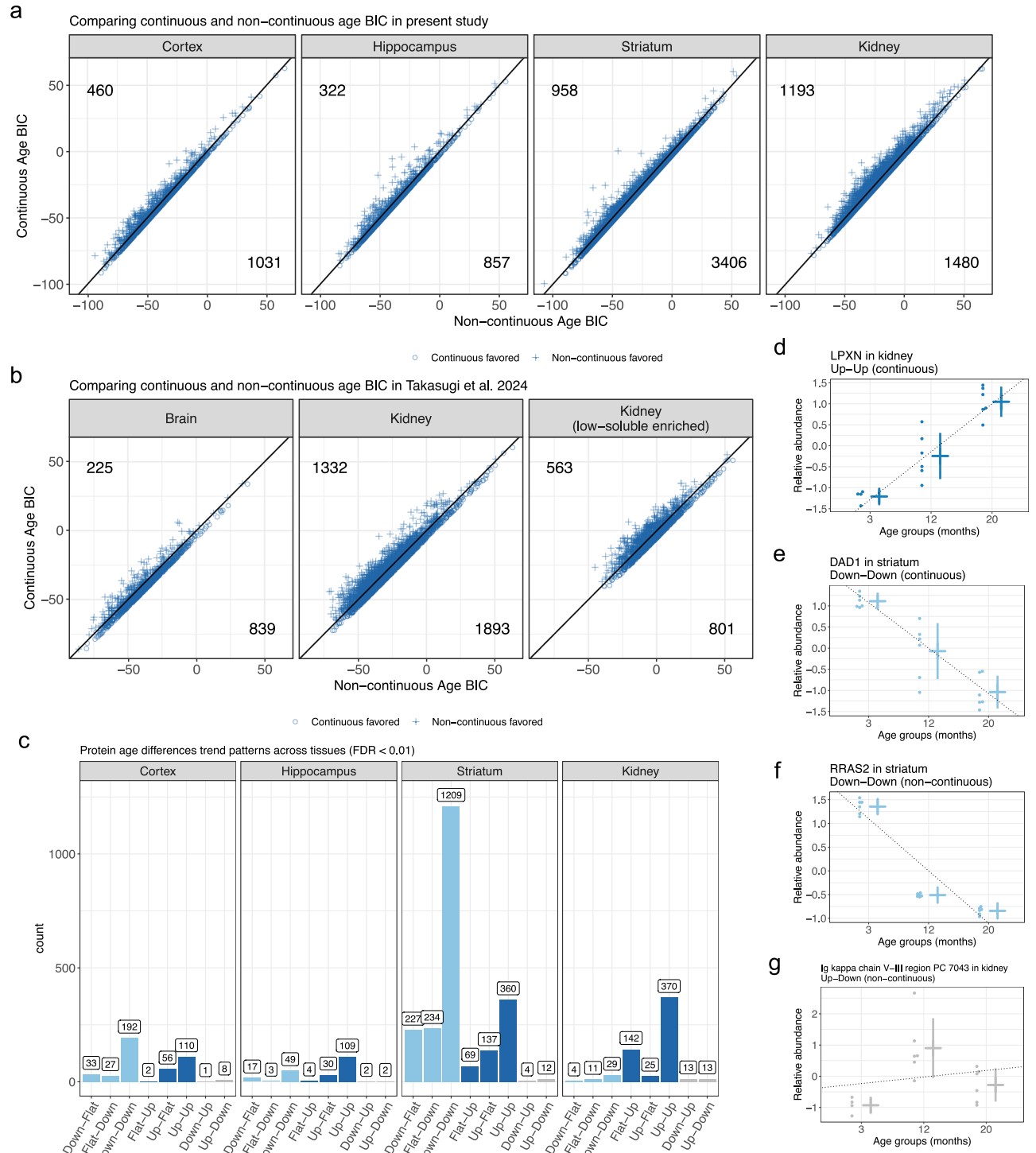

**Fig. 4 | Proteins with distinct aging trends observed across tissues. a** BIC comparison of continuous and non-continuous model fits of the age term for significant age differences (FDR < 0.1). Age differences in individual proteins tested with F tests, followed by FDR adjustment. Counts of proteins more consistent with a continuous age model are in the bottom right and a non-continuous model are in the top left. **b** BIC comparison of continuous and non-continuous model fits of the age term for significant age differences (FDR < 0.1) in Takasugi et al.[14] data. Age differences in individual proteins tested with F tests, followed by FDR adjustment. Counts of proteins more consistent with a continuous age model are in the bottom right and a non-continuous model are in the top left. **c** Categorization of highly significant (FDR < 0.01) age trends as stepwise changes between age groups. Age differences in individual proteins tested with F tests, followed by FDR adjustment. **d** LPXN was more consistent with an upward continuous trend with age in kidney.

N = 4 biological replicates for 3-month group and N = 6 for 12- and 20-month groups. Mean ± SD indicated for each age group. Dotted best fit line included for reference. **e** DAD1 was more consistent with a downward continuous trend with age in striatum. N = 6 biological replicates for all age groups. Mean ± SD indicated for each age group. Dotted best fit line included for reference. **f** RRAS2 was more consistent with a non-continuous downward trend with age in striatum. N = 6 biological replicates for all age groups. Mean ± SD indicated for each age group. Dotted best fit line included for reference. **g** Ig kappa chain V-III PC 7043 was more consistent with a non-continuous aging differences in kidney, which had an Up-Down trend with age. N = 4 biological replicates for 3-month group and N = 6 for 12- and 20-month groups. Mean ± SD indicated for each age group. Dotted best fit line included for reference.

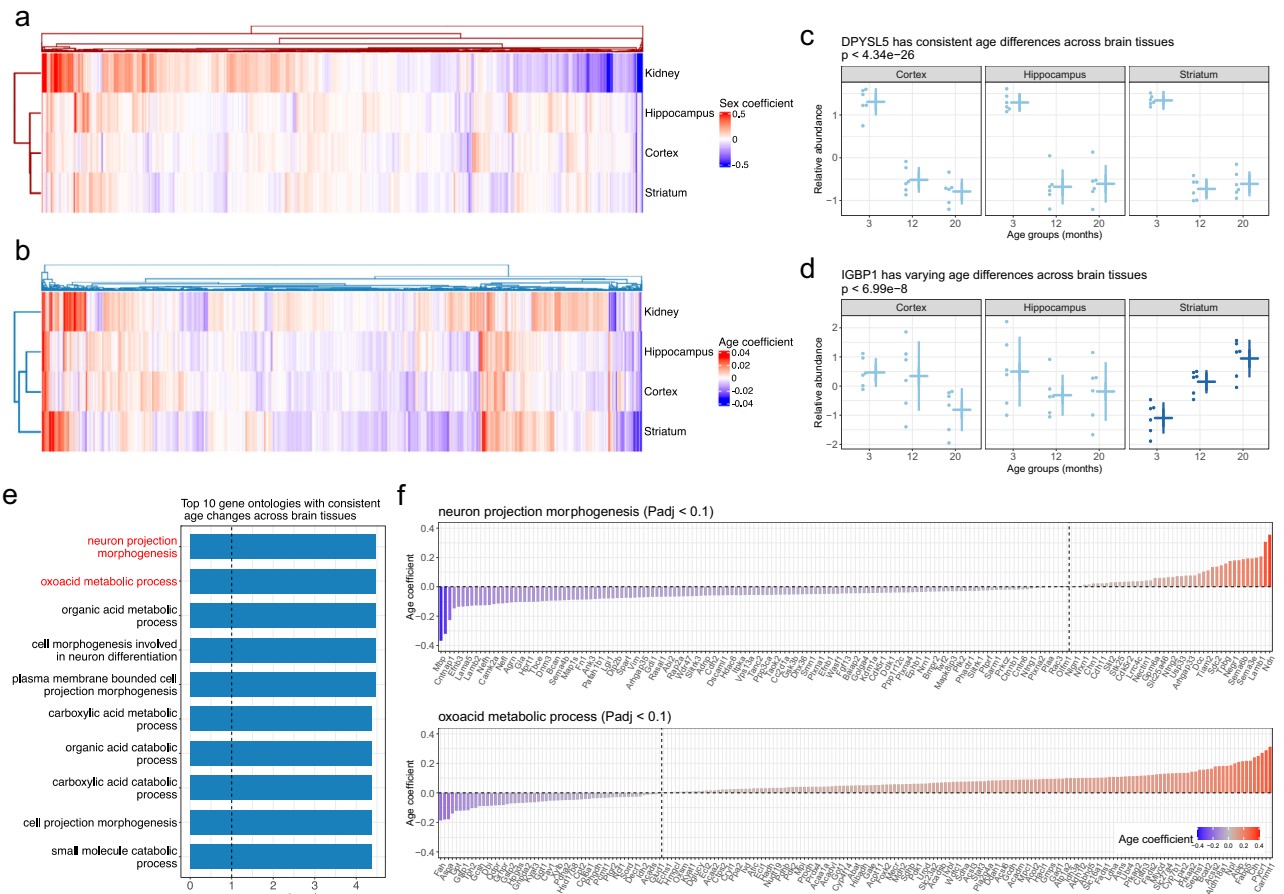

**Fig. 5 | Cross-tissue modeling of age and sex differences. a** Heatmap of sex coefficients for 5778 proteins observed in all four tissues. **b** Heatmap of continuous age coefficients for 5778 proteins observed in all four tissues. **c** DPYSL5 had consistent age differences across the three brain tissues. N = 6 biological replicates for all age groups in each tissue. Mean ± SD indicated for each age group in each tissue. F test used to test for consistent age differences across tissues. **d** IGBP1 had inconsistent age differences across the three brain tissues. N = 6 biological replicates for all age groups in each tissue. Mean ± SD indicated for each age group in each tissue. F test used to test for age-by-tissue differences across tissues. **e** Top 10

gene ontologies enriched in proteins with consistent age differences across the three brain tissues. Enrichment analysis performed using hypergeometric tests, followed by FDR adjustment. Vertical dashed line represents a threshold of FDR < 0.1. Neuron projection morphology and oxoacid metabolic process are highlighted with red text, which are featured in Fig. 5f. **f** Proteins with significant consistent age differences across the three brain tissues (FDR < 0.1) in the neuron projection morphology (top) and the oxoacid metabolic process (bottom) gene ontologies. Vertical dashed lines divide positive and negative age coefficients. Every other protein included as a label for clarity.

ones. We filtered to the 7377 proteins detected in all three brain tissues and tested for age-related differences, as well as age-by-tissue interactions (Methods, Supplementary Data 8). This analysis allowed us to pinpoint proteins with either uniform age trends or tissue-specific variation (Methods).

This analysis also enabled us to identify proteins with consistent age trends across brain tissues, as well as those with tissue-specific variation. For example, we identified DPYSL5, which displayed a consistent non-continuous age trend across all brain tissues, and IGBP1, which exhibited an increasing continuous age trend specific to the striatum (Fig. 5c, d).

We performed GSEA[33] based on the set of 2245 genes with evidence for consistent age differences (marginal age $P_{adj}$ < 0.1 and age-by-tissue $P_{adj}$ > 0.2) to identify gene ontologies associated with shared aging patterns across the brain. We observed overarching patterns of proteins related to metabolism (*e.g.*, oxoacid metabolic process) increasing with age and neurons and synapses (*e.g.*, neuron projection morphogenesis) decreasing with age (Fig. 5e, f). The observed negative aging trends for proteins involved in synapse biology aligns with previously reported findings by Tsumagari et al.[10], which profiled the cortex and hippocampus proteomes of male mice from another C57BL/6 sub-strain (C57BL/6 J Jcl) in similar age groups (3, 15, and

24 months-old). Notably, we confirmed this trend in both male and female mice in our study.

**Comparing protein age dynamics between adulthood and early development**

While most age-related differences appeared continuous across the span of adulthood studied here, some proteins exhibited non-continuous trends. For instance, AKR1C13 showed a decreasing trend in females during middle-to-late adulthood (Fig. 3f). However, this non-continuous pattern became apparent only when young adult female mice were included in the analysis. This observation led us to explore how protein age differences during adulthood in our study compare to changes observed in early life development. To address this, we integrated our findings with early developmental data from Wang et al.[11]. This complementary study (Fig. 6a) profiled the proteomes of 10 tissues, including whole brain and kidney, in both female and male C57BL/6 J mice across infancy-to-adolescent stages (1 week, 1 month, and 2 months). To compare between the studies, we re-analyzed the brain and kidney data from Wang et al.[11] using a consistent statistical approach to minimize discrepancies due to analysis (Methods). Fundamental differences remain between the studies (*e.g.*, different institutions at different times, sister sub-strains of mice,

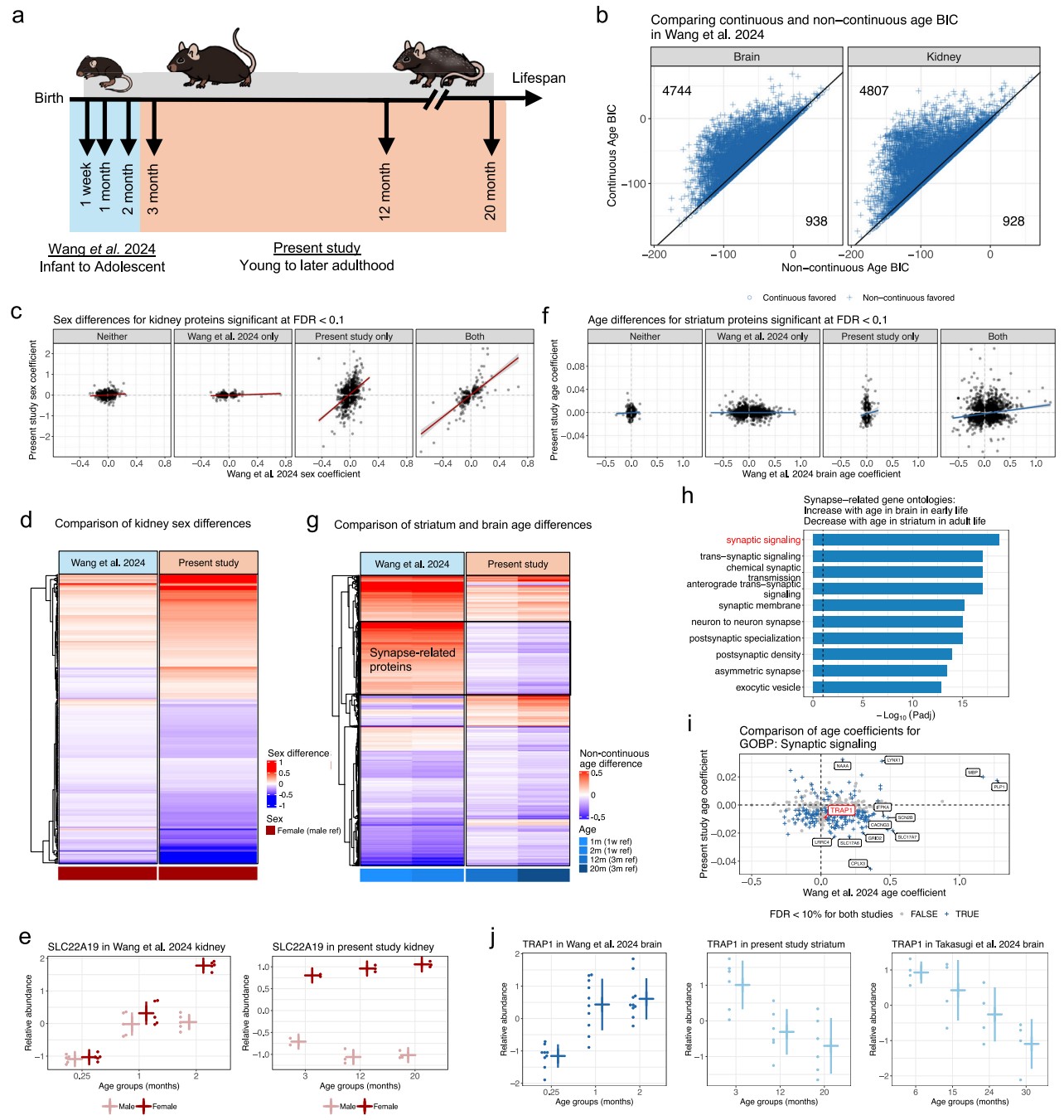

differences in tissue collection, and DIA vs TMT proteomics). PCA revealed similarities and differences in the distribution of variation across the the present study, Takasugi et al.[14], and Wang et al.[11]. In Wang et al.[11], the first three principal components accounted for 91.8% of the total variation (Supplementary Fig. 7a), compared to 97.5% (Supplementary Fig. 4d) and 98.2% (Supplementary Fig. 7d) for the adulthood studies. Despite this, the contributions of tissue, age, and sex to variation differed slightly among the studies. In Wang et al.[11], PC1 primarily separated brain and kidney, while PC2 distinctly separated age groups (Supplementary Fig. 7b), with the youngest age group (1 week) showing a marked separation from the older age groups (1 and 2 months). By contrast, in the adult studies, age variation appeared more continuous (Supplementary Fig. 4f and Supplementary Fig. 7e). On the other hand, sex strongly correlated with PC5 (Supplementary Fig. 7c). Interestingly, compared to adulthood in the present study, age explained a larger proportion of overall variation during early

development, whereas sex variation was more prominent in adulthood. Additionally, age-related changes during early development appeared distinctly less continuous (Fig. 6b) compared to age trends observed in adulthood (Fig. 4a, b). Taken together, this suggests protein levels have more extreme changes early in development (between 1 week and 1 month) which become more linear during adolescence and adulthood.

### Protein sex differences in kidney become more pronounced in adulthood

We next compared age and sex difference coefficients between the two studies. In the kidney, we observed weak concordance for age coefficients (Supplementary Fig. 8a, b) but strong concordance for sex coefficients between early development and adulthood, particularly if the difference was statistically significant (FDR < 10%) in both studies (Fig. 6c, d). Interestingly, sex differences that were only statistically

**Fig. 6 | Comparison to early development protein data reveals distinct aging patterns from adulthood. a** Diagram of the age groups observed across Wang et al.[11] and present study. **b** BIC comparison of continuous and non-continuous model fits of the age term for significant age differences (FDR < 0.1). Age differences in individual proteins tested with F tests, followed by FDR adjustment. Counts of proteins more consistent with a continuous age model are in the bottom right and a non-continuous model are in the top left. **c** Comparison of sex coefficients for kidney between the present study and Wang et al.[11]. Coefficients stratified into groups based on which studies detected a significant difference (FDR < 0.1). Best fit line from regressing sex coefficients from the present study on those from Wang et al.[11] ± SE included for reference. Sex differences in individual proteins tested with F tests, followed by FDR adjustment in each study. **d** Heatmap of sex coefficients for kidney proteins that were significant (FDR < 0.1) in both present study and Wang et al.[11]. Sex differences in individual proteins tested with F tests, followed by FDR adjustment in each study. **e** SLC22A19 exhibited a sex difference in kidney that became more pronounced with age, characterized by higher levels in females. The trend is observed across early development (left) and throughout adulthood (right). For Wang et al.[11], N = 5 biological replicates for all age-by-sex groups. For present study, N = 2 biological replicates for both 3-month groups and N = 3 for 12- and 20-month groups. Mean ± SD indicated for each age-by-sex group. **f** Comparison of age coefficients between the present study's striatum tissue and Wang et al.[11]'s brain tissue. Coefficients are stratified into groups based on which studies detected a significant difference (FDR < 0.1). Best fit line from regressing

age coefficients from the present study on those from Wang et al.[11] ± SE included for reference. Age differences in individual proteins tested with F tests, followed by FDR adjustment in each study. **g** Heatmap of non-continuous age coefficients that were significant (FDR < 0.1) in both present study striatum proteins and Wang et al.[11] brain proteins. Age differences in individual proteins tested with F tests, followed by FDR adjustment in each study. Black box highlights proteins with increasing trends with age in early development and decreasing trends in adulthood, which are enriched in synapse-related functions. **h** Top 10 gene ontologies enriched in proteins with upward age trends in brain tissue during early development and downward age trends in striatum through adulthood. Enrichment analysis performed using hypergeometric tests, followed by FDR adjustment. Vertical dashed line represents a threshold of FDR < 0.1. Synaptic signaling is highlighted with red text, which is featured in Fig. 6i. **i** Comparison of age coefficients between the present study's striatum tissue and Wang et al.[11]'s brain tissue for proteins in the synaptic signaling gene ontology. Vertical and horizontal dashed lines included for reference, highlighting proteins with discordant age coefficients between the two studies. TRAP1 is highlighted with red text, which is featured in Fig. 6j. **j** TRAP1 exhibited an upward trend with age in the brain during early development into adolescence (left). This trend reversed in adulthood (middle) and even into geriatric age at 30 months-old (right). For Wang et al.[11], N = 9 biological replicates for 1-week group and N = 10 biological replicates for 1- and 2-month groups. For present study, N = 6 biological replicates for all age groups. For Takasugi et al.[14], N = 4 biological replicates for all age groups. Mean ± SD indicated for each age group.

detected in adulthood data showed moderate concordance with early development data, while differences unique to the earlier developmental data exhibited much lower concordance. These findings suggest that many sex differences in adulthood are initiated early in development but become more pronounced in adulthood, potentially coinciding with sexual maturity. One example is SLC22A19 (Fig. 6e), an organic anion transporter known to have higher expression in females[39], which does not exhibit this pattern until 2 months and then becomes more pronounced later in adulthood (~12 months-old). In the brain, sex differences were relatively sparse in both studies. Moreover, those identified differences were largely inconsistent in direction between early development and adulthood (Supplementary Fig. 8c), suggesting a more complex or variable progression of sex differences in brain tissues across the lifespan.

### Complex age dynamics of synapse-related proteins throughout lifespan in brain tissues

Next we compared the Wang et al.[11] brain data to all three brain tissues (cortex, hippocampus, and striatum) in the present study (striatum in Fig. 6f; all brain tissues in Supplementary Fig. 8d). We note that the continuous age coefficient is a less accurate summary for early development given the extensive non-continuous age differences observed. Notably, far more age differences were detected in the early developing brain, and these did not strongly correlate with adult brain age differences. Even when a protein's age difference was statistically significant in both studies (FDR < 10%), many showed discordant patterns, with the direction of change often reversed between early development and adulthood (Fig. 6g for striatum). One notable pattern within proteins exhibiting a potential reversal over the lifespan—specifically an increase in abundance during early development paired with a decrease in adulthood—was enriched for genes associated with synapse biology (Fig. 6h, i). This observation is highly consistent with synapse density across lifespan as measured through deep synaptome profiling[40]. To incorporate geriatric mouse data, we further filtered to brain proteins with both an upward aging trend in Wang et al. [11] and downward trends in the present study and Takasugi et al. [14], particularly in the geriatric age group (30 months-old). This highlighted SHISA7, an AMPA receptor-associated protein that regulates hippocampal synapses and contextual memory[41], and TRAP1, a mitochondrial chaperone that is down-regulated in Alzheimer's Disease[42]. Integration and validation across aging data resources can elucidate

developmental networks in the aging brain that represent candidates for treatment to counteract age-related structural changes and functional decline.

## Discussion

In this study, we integrated the Orbitrap Astral Mass Spectrometer with TMT technology for a multiplexed proteomic analysis. This approach enabled us to overcome previous limitations in proteome coverage and depth and boost the quantified protein number significantly, yielding insights into tissue-specific, age-related, and sex-based protein abundance patterns. During our analysis, we discovered that ion saturation in the Astral analyzer during TMT reporter ion quantitation can lead to inaccuracies, potentially resulting in the false discovery of biological effects. This issue is specific to the TMT and Orbitrap Astral combination and does not affect other DDA or DIA applications using other mass spectrometers. It can be mitigated by injecting less material; however, this approach may reduce the depth of protein quantitation. To address the challenge, we developed and applied a stringent peptide filtering strategy based on resolution and signal-to-noise (S/N) thresholds specifically tailored for the TMT datasets generated on the Orbitrap Astral system. Although the filtering process resulted in a 20% reduction in the total number of peptides and a 15% decrease in unique peptides, the impact on protein quantitation was minimal, with only a ~5% reduction in the total number of quantified proteins (Fig. 2c). However, the filtering process influenced a significant proportion of proteins. Resolution-based filtering affected more than one-third of all quantified proteins, while saturated ions impacted over 20%, and S/N filtering impacted ~80% of the quantified proteins (Fig. 2d). Despite these reductions, the filtering strategy effectively mitigated biases associated with low-abundance and low-resolution peptides. This improvement in data quality enhanced the accuracy of protein quantitation, enabling us to capture complex molecular changes linked to aging with greater confidence. Our strategy provides the proteomics community with a robust approach for analyzing TMT datasets generated with the Orbitrap Astral Mass Spectrometer. Additionally, we recommend performing weekly Astral resolution checks and evaluating the resolution of TMT reporter ions acquired with the Orbitrap Astral Mass Spectrometer before proceeding with further data analysis to ensure optimal data accuracy and reliability. For TMT experiments with limited injection material, such as single-cell proteomics, we also recommend checking

the resolutions of the TMT reporter ions, given the substantial dynamic range of protein abundance in cellular and tissue samples.

We profiled both sexes across three distinct age groups (3, 12, and 20 months) and provided a comprehensive view of proteomic changes from young adulthood to early late life. We observed distinct tissue-based proteomic trends, notably the sex differences in the kidney as contrast to the minimal sex influence in brain tissues[15]. This finding aligns with the known susceptibility of the kidney to sex-linked aging effects. Compared to previous studies, the wider dynamic range of ages included in this study improved statistical power and enabled the identification of more proteins with sex and age differences (Supplementary Fig. 6).

From a discovery perspective, this study also identified compelling examples that can be further investigated to elucidate underlying aging mechanisms. One such example is AKR1C13, which exhibits a non-monotonic up-down pattern in female mice, a trend that only becomes apparent with our broader age range (Fig. 3f). AKR1C13 is an NADP(H)-dependent oxidoreductases belonging to the aldo-keto reductase (AKR) family[43]; It has broad substrate specificity for 20α-, 17β- and 3α-hydroxysteroids, and non-steroidal alcohols[44]. The increased level of AKR1C13 in adult female mice may indicate a heightened demand for lipid metabolism. In contrast, the decreased level of AKR1C13 in aged female mice could contribute to the accumulation of lipid peroxidation products, elevated reactive oxygen species (ROS), and increased oxidative damage[45]. This study functions as a quantitative protein resource for the aging-focused research community. We have made our data and processed results available to the aging research community, accessible online. It allows for querying of proteins of interest, facilitating deeper exploration of the dataset. Results can also be integrated with other complementary studies[11,14] to produce higher confidence candidates for deeper interrogation.

Prior aging studies have revealed complex non-continuous transcriptional aging trends, such as Schaum et al.[5], which profiled 10 age groups. Having three adult age groups in the present study, four adult age groups from Takasugi et al.[14], and three early development age groups from Wang et al.[11], allowed us to more deeply characterize age trends than previous aging proteomic studies. We observed the presence of non-continuous age-related protein trajectories, emphasizing the complexity of aging at the molecular level. Expanding the dynamic range of age within a study to broadly encompass the entire murine lifespan would enable more complex models that capture highly non-continuous dynamics. Based on three age groups per sample population, the minimum number of groups to assess deviation from a continuous trend, our comparisons of developmental and adulthood aging dynamics revealed at times opposite trajectories, such as for proteins involved in synaptic functions. This highlighted that the aging process not only involves a linear accumulation of changes but also complex reversals and non-linear patterns across the lifespan. Our power to capture these patterns is limited when comparing independent studies and motivates future studies with more age groups and tissues within a single study.

While this study provides valuable insights into age-associated proteomic changes, it also has some limitations. First, our analysis was limited to four tissues. Expanding the study to a broader range of tissues could provide a more comprehensive view of proteomic aging processes across the body. Second, we applied a stringent cutoff to filter low-intensity ions, which, while necessary for ensuring quantitation accuracy, may have excluded biologically relevant peptides. This approach might result in a minor loss of depth and could overlook subtle yet meaningful proteomic variations that contribute to the aging process. This study focused primarily on protein abundance changes. Including post-translational modification (PTM) analysis in future research would offer a deeper regulatory perspective on aging, providing insights into protein functionality and interaction that abundance data alone cannot

capture. Future studies addressing these limitations could build on these findings to achieve a more integrated and translatable understanding of proteomic dynamics in the process of aging. In terms of statistical analysis, comparing findings across studies is always subject to the factors that differ among the studies. Our statistical approach using summary statistics rather than full data integration help mitigate potential issues, but it is important to keep in mind that these studies survey different sub-strains of C57BL/6, encompass different age groups, and were performed at different institutions using differing mass spectrometry approaches.

Overall, this work provides a robust analysis framework for TMT datasets generated using Orbitrap Astral mass spectrometer, illustrating the potential of this advanced proteomic technology for mapping age-related molecular alterations across tissues. We expanded the proteomic landscape of aging by increasing the extent of protein quantitation across a wider range of lifespan, highlighting the need for further research to explore these age-related protein dynamics, particularly the roles of synaptic proteins in the aging processes and age-associated disease conditions.

## Methods
### Tissue collection
All animal protocols were approved by the Institutional Animal Care and Use Committee at Sanford Burnham Prebys Medical Discovery Institute (AUF 23-084 and AUF 24-069) and were conducted in compliance with all relevant ethical regulations. C57BL/6JN mice were ordered through the National Institute of Aging (NIA). The mice were group housed at 21–24 °C, 30–70% humidity, and a 12 hr light/dark cycle under specific pathogen-free conditions with *ad libitum* access to water and food. Upon arrival, mice were acclimated to the local facility for one month. To collect tissues, mice were euthanized by cervical dislocation, and tissues were collected as quickly as possible. Brain tissues were dissected on filter papers soaked with ice-cold PBS under a dissection microscope. Each brain region was flash-frozen in liquid nitrogen upon dissection. Kidney tissues were dissected and flash-frozen in liquid nitrogen. Three replicates were collected from both female and male mice at the age of 3, 12, and 20 months.

### Sample preparation for proteomic analysis
Tissue samples were homogenized using the FastPrep-24 Instrument using lysing matrix D in lysis buffer (8 M urea, 200 mM EPPS, Roche protease inhibitor tablets) and cells sonicated. Lysates were cleared by centrifugation (10 min at 13,000 g at 4 °C) and protein concentrations were measured using Pierce BCA assay kits. Proteins were then reduced with TCEP (5 mM for 15 min at room temperature) and alkylated with iodoacetamide (15 mM for 15 min in the dark). The alkylation reaction was quenched by DTT (10 mM for 15 min). For each sample, 25 μg of protein was aliquoted and diluted to a final concentration of 1 mg/mL and cleaned by SP3 procedure[46]. Proteins were digested using LysC (Wako, 3 hr, 37 °C, 1,400 rpm on a ThermoMixer) followed by trypsin (6 hr, 37 °C, 1400 rpm). The resulting peptides were then labeled with TMTpro 18-plex reagents (Thermo Fisher Scientific) for 1 hr at room temperature. The reaction was quenched with hydroxylamine (final concentration of 0.5% for 15 min). Labeled peptides were mixed. After labeling and mixing, peptide mixtures were desalted using C18 seppak cartridges (1 mg, Waters). Desalted peptides were then fractionated using basic-pH reverse phase chromatography[17]. Briefly, peptides were resuspended in Buffer A (10 mM ammonium bicarbonate, 5% acetonitrile, pH 8) and separated on a linear gradient from 13% to 42% Buffer B (10 mM ammonium bicarbonate, 90% acetonitrile, pH 8) over an Agilent 300Extend C18 column using an Agilent 1260 HPLC equipped with single wavelength detection at 220 nm). Fractionated peptides were desalted using Stage-tips[17] prior to LC-MS/MS analysis.

## Mass spectrometry data acquisition

Peptides were separated prior to MS/MS analysis using a Neo Vanquish (Thermo Fisher Scientific) equipped with an in-house pulled fused silica capillary column with integrated emitter packed with Accucore C18 media (Thermo). Separation was carried out with 75-minute gradients from 96% Buffer A (5% ACN, 0.125% formic acid) to 30% Buffer B (90% ACN, 0.125% formic acid). Mass spectrometric analysis was carried out on an Orbitrap Astral Mass Spectrometer (Thermo). Peptides and proteins were filtered to 1% using the rules of protein parsimony[47]. For MS1, Orbitrap resolution was set to 60k, and the normalized AGC Target was set to 200%. The purity threshold was set to 70%, and purity window is 0.5%. For MS2, the isolation window was set to 0.5 m/z. HCD collision energies were set to 35%, with a maximum injection time of 20 ms. FAIMS voltages of −30, −35, −45, −55, −60 were used.

## Mass spectrometry data analysis

Spectra were converted to mzXML using a modified version of ReAdW.exe. Database searching included all entries from the *Mus musculus* UniProt database (January 26, 2022). This database was concatenated with one composed of all protein sequences in the reversed order. Searches were performed using a 50-ppm precursor ion tolerance for total protein level profiling. The product ion tolerance was set to 0.03 Da. These wide mass tolerance windows were chosen to maximize sensitivity in conjunction with COMET searches and linear discriminant analysis[48,49]. TMT tags on lysine residues and peptide N termini (+304.2071 Da) and carbamidomethylation of cysteine residues (+57.021 Da) were set as static modifications, while oxidation of methionine residues (+15.995 Da) was set as a variable modification. Peptide-spectrum matches (PSMs) were adjusted to a 1% false discovery rate (FDR)[50,51]. PSM filtering was performed using a linear discriminant analysis, as described previously[48] and then assembled further to a final protein-level FDR of 1%[50]. TMT reporter quantitation was performed using both 0.001 Da and 0.003 Da peak match tolerance (PTM) windows. The dataset using the 0.001 Da PTM was used for the subsequent analysis. 0.001 Da Peptide-Spectrum Matches (PSM) with any of the 18 reporter ions resolutions lower than 45 K and a total signal-to-noise (S/N) ratio less than 1440 across 18 channels were filtered out. Proteins were quantified by summing reporter ion counts across all matching PSMs, as described previously[28]. The S/N measurements of peptides assigned to each protein were summed and these values were normalized so that the sum of the signal for all proteins in each channel was equivalent to account for equal protein loading.

## Peptide filtration and summarization to proteins

Peptides that did not have full resolution and/or S/N lower than 1440 for 18 samples were removed and not used for protein analysis. Abundance levels for proteins (based on UniProt ID) were estimated by summing their retained peptides. The 18-plex TMT study design simplified data processing and normalization because all samples per tissue could be analyzed within a single batch and thus not requiring a batch adjustment.

## Detecting age- and sex-related differences for proteins in individual tissues

For a given tissue, we tested for proteins that differed across age groups, fit either as continuous or non-continuous (categorical), sexes, and age-by-sex groups. To test age differences, we fit the following log-linear regression model for each protein:

$$\log_2(\text{protein}) = \text{intercept} + \text{age} + \text{sex} \qquad (1)$$

where age represents either a continuous covariate (in terms of months), and thus the model estimates only a single coefficient or a non-continuous categorical covariate factor, fit as two coefficients.

The age difference was then tested with an F-test by comparing to a null model excluding the age term. Sex was tested similarly, except with the null model excluding the sex term instead. To test for age-by-sex differences, we fit the following regression model:

$$\log_2(\text{protein}) = \text{intercept} + \text{age} + \text{sex} + \text{age-by-sex} \qquad (2)$$

where age-by-sex represents the interaction effect between age and sex (with age fit as either a continuous or non-continuous term). The interaction term was then tested using an F-test, comparing the model in Eq. 2 to a null model excluding the interaction term (Eq. 1). The p-values were then FDR-adjusted using the Benjamini-Hochberg method[52].

We also performed these tests for individual peptide level data to assess trends for how resolution and S/N affected coefficient estimation when compared to protein-level coefficients (Supplementary Fig. 2). To assess whether the age differences were more consistent with a continuous fit or non-continuous fit, we compared the non-nested models using the Bayesian Information Criterion (BIC), a model selection criterion. A lower BIC corresponds to the BIC-supported model.

## Detecting consistent and distinct age- and sex-related differences for proteins across brain tissues

We jointly modeled tissues together to detect age differences that were consistent across tissues, representing the main effect of age, as well as allowing an age-by-tissue interaction to capture tissue-distinct differences. We expanded the model in Eq. 1 to:

$$\log_2(\text{protein}) = \text{intercept} + \text{age} + \text{sex} + \text{tissue} + \text{age-by-tissue} \qquad (3)$$

where the model now includes a main effect for tissue and its interaction with age. We tested for the age main effect with an F test comparing Eq. 3 to a null model fit excluding the age term. We also tested for an age-by-tissue effect with an F test comparing Eq. 3 to a null model fit excluding the interaction term. The same approach was also used to test for a sex main effect and sex-by-tissue interaction across tissues. The p-values were then FDR-adjusted using the Benjamini-Hochberg method[52].

## Principal component analysis (PCA)

PCA was performed using the pcaMethods R package[53]. We combined data across all tissues, representing 70 samples, after filtering to the 5778 proteins observed in all four tissues. We first $\log_2$ transformed each protein to account for an expected overall log-linear distribution. We estimated 10 principal components (PCs).

## Gene set enrichment analysis (GSEA)

GSEA was performed using the clusterProfiler R package[33]. Age- and sex-differences were used to define gene sets (*e.g.*, age coefficient > 0 and $P_{\text{adj}} < 0.1$ for a given tissue), which were then compared to a universe gene set (*e.g.*, all analyzed proteins for a given tissue). Enriched gene ontologies and KEGG pathways were detected based on an FDR threshold, such as $P_{\text{adj}} < 0.1$. We also defined gene sets based on stepwise categorical age trends (*e.g.*, "Down-Flat" for a protein that significantly dropped in abundance between 3 and 12 months but did not change between 12 and 18 months). We were more stringent in defining age trend gene sets (FDR < 1%) given the greater potential to over-fit the data. The unique gene identifiers were ENSEMBL gene IDs for gene ontology analysis and UniProt IDs for KEGG pathway analysis.

## Analysis of kidney and brain proteomic data from Takasugi et al.[14]

We re-analyzed the kidney and brain tissue data from Takasugi et al.[14] to ensure consistent modeling with the present study. It was a cross-

sectional study of adulthood-to-geriatric ages in 8 tissues (aorta, brain, heart, kidney, liver, lung, muscle, and skin). Both whole tissue lysates and low-solubility protein-enriched fraction samples were run for many of the tissues. Four male C57BL/6 J mice were collected from four age groups: 6 months, 15 months, 24 months, and 30 months. TMT 16-plex on a Lumos mass spectrometer was used compared to the current study's TMT 18-plex on an Astral mass spectrometer. We used the same statistical modeling approach as for the current study, excluding any covariate adjustment for sex or testing for sex-related differences and including both continuous and non-continuous age modeling (Takasugi et al.[14] reported non-continuous results). The non-continuous model was extended to include four age groups. We also performed PCA as described for the current study.

**Analysis of kidney and brain proteomic data from Wang et al.[11]**
We re-analyzed the kidney and brain tissue data from Wang et al.[11] to ensure consistent modeling with the present study. It was a cross-sectional study of early life development in 10 tissues (brain, heart, intestine, kidney, liver, lung, muscle, skin, spleen, and stomach). Five C57BL/6 mice (from Shanghai Jiesijie animal Co., Ltd) of each sex were collected from three age groups: 1 week, 1 month, and 2 months. Data-independent acquisition (DIA) mass spectrometry was used instead of data-dependent acquisition (DDA) (*e.g.*, the current study's TMT 18-plex approach). After filtering out proteins with > 80% missing values for a tissue, we used the same statistical modeling approach as for the age and sex differences as with the current study, including both continuous and non-continuous age modeling (Wang et al.[11] used non-continuous modeling in their reported results[11]). We also performed PCA and GSEA as described for the current study.

All the data analysis in the Method section were performed in R (v4.4.3), and the R packages used include tidyverse (v2.0.0), readxl (v1.4.4), writexl (v1.5.2), clusterProfiler (v4.14.6), DOSE (v4.0.0), ggdist (v3.3.2), ComplexHeatmap (v2.22.0), circlize (v0.4.16), ggrepel (v0.9.6), UpSetR (v1.4.0), pcaMethods (v1.98.0) and ggsci (v3.2.0).

**Reporting summary**
Further information on research design is available in the Nature Portfolio Reporting Summary linked to this article.

## Data availability
The mass spectrometry proteomics data generated in this study have been deposited to the ProteomeXchange Consortium via the PRIDE[54] partner repository with the dataset identifier PXD058160.

## Code availability
All data processing and statistical analyzes were conducted with the R statistical programming language[55]. All processed data, including supplementary tables, and the R code used for statistical analyzes and generating the reported findings are publicly available on Figshare (https://doi.org/10.6084/m9.figshare.28016501). README file at the top of the directory provides a description of directory contents, which includes an R script for generating each figure.

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

## Acknowledgements

We thank all the Zhang Lab members for active discussion and support. T.Z. was supported by NCI grant R00CA273170. We thank the SBP animal facility, particularly Adriana Charbono and Feliciano Lopez, for managing the mouse colonies. J.A.P. was supported by NIGMS grant R35GM156406. X.T. was supported by NIA grant R00AG068303, NCI grant P30CA030199, Longevity Impetus Grant from Norn Group, Hevolution Foundation and Rosenkranz Foundation. We thank Corinne M. Keele for her illustrations of a B6 mouse across its lifespan (Fig. 6a).

## Author contributions

X.T. and T.Z. conceptualized the project and designed the experiments. Y.D., S.P.K., and T.Z. performed the experiments. E.D.J., J.A.P., S.P.G., and T.Z. designed the methodology. G.R.K. designed the statistical approach. E.H. and Z.G. harvested the mouse tissues. G.R.K., D.L.B., and T.Z. performed the analysis. G.R.K., J.A.P., S.P.G., X.T., and T.Z. wrote the manuscript. All authors reviewed the manuscript.

## Competing interests

The authors declare no competing interests.
