## [Transparent Peer Review file · Nature Communications]

Expanding the Landscape of Aging via Orbitrap Astral Mass Spectrometry and Tandem Mass Tag Integration

Corresponding Author: Dr Tian Zhang

Version 0:

Reviewer comments:

Reviewer #1

(Remarks to the Author)

In their study, Keele and colleagues investigate brain development during aging, considering sex and brain-region differences in mice, thereby enhancing our understanding of aging processes. They compare their findings to a previous study focusing on infant mice. Notably, their work represents the first published study to perform isobaric multiplexing at the 18plex level using the new Orbitrap Astral MS, which will undoubtedly be of highest interest for the community from a technical and methodological perspective. As I am not a biologist, I cannot comment on the significance for research in brain development and aging, but the findings appear exciting to me.

While the authors discuss some existing limitations of their study, I have additional questions and comments that should be addressed prior to publication:

1. Figure 1B: After counting, it appears that 18 signals are annotated in the spectrum, but not all correlate to the 18 expected reporter ions (RI). If I understand correctly, the authors describe detecting only 12 signals. I suggest aligning the text with the figure and annotating the visible RI signals, as some seem not fully resolved in the given example.
2. Resolution Issue: Likely caused by ion saturation, can the authors comment on the feasibility of using their TMT workflow for (ultra) low input samples, such as single cells or small laser microdissections of tissue where sample input is limited? I would assume fewer to no cases of ion saturation would be observed in such cases, while the extremely high sensitivity of the Astral MS would be highly beneficial (in addition to its speed) compared to Orbitrap-based TMTPro workflows. Have the authors tested lower inputs? If not, is it possible to test this (e.g., with sample leftovers that could be diluted)? If done, it would significantly enhance the manuscript's quality by opening a new use-case scenario for their workflow, even if it is just a proof-of-principle test showcasing the minimal input needed to obtain meaningful data.
3. Figure 2A & 2B: If I understand correctly, the main message is that RI from low and high-intensity peptides have a resolution problem, and filtering is needed for accurate quantification. However, this was hard to understand from the figure, and the figure legend reads more like the results section rather than a figure description. For example, log S/N was plotted according to the legend, but neither the x nor y axis is annotated with log S/N, which seems confusing at first glance. I recommend making the nomenclature in the legend and text consistent. Also, high S/N is used in Fig2D, but SN>1440 is used in 2B – I recommend clearly indicating if the same thing or something different is meant by “high.”
4. Figure 2D: This figure would benefit from better annotation. For example, instead of “count,” use something like “# proteins quantified” as axis annotation, or add a header as done in 2C, to make it easier to understand the figure at first glance. This also applies to Figure 2D/line 285 and following: It was not obvious to this reviewer what was filtered (shown in light blue) if not S/N & R. It was also hard to understand the difference from 2C/proteins. I assume “impact” here means different regulation of proteins that are still quantified? Please clarify the figure's appearance here.
5. Sample Replicates: If I understand correctly, samples were collected in triplicates for each condition. If so, please indicate what numbers are represented by the bars in Figure 2. Are they averages from n=3? If so, I recommend adding error bars and indicating the number of replicates used in the figure legend.
6. Figure S5: I appreciated the comparison to the previous study by the same first author, recorded on a Lumos. Can the authors include a comment on the overall protein numbers and overlap found in their previous study compared to this one?

How significant is the advantage they see on the Astral over the Lumos, given the need for filtering on the Astral? This information would be valuable for the community in deciding which instrument type might be best suited for their TMT project.

7. Workflow Comparison: In line with the value of instrument comparison discussed above, can the authors comment on the advantages and limitations they see within their TMT on Astral versus an LFQ-DIA on Astral approach, as performed by the cited Wang et al. study? The authors mention in their discussion: "This approach enabled us to overcome previous limitations in proteome coverage and depth and significantly boost the quantified protein number." However, it is not fully clear to the reader which competing strategy this boost was compared to. Please accept my apologies if I missed this in the text. A direct comparison of TMT vs. DIA would be helpful for the reader to make an informed choice before starting a new study. This includes estimated coverage or required measurement time (I think the authors generated 96 fractions from their TMT mix, pooled to 24 fractions, of which 12 were measured; how would the required MS time compare to direct measurements in DIA mode without fractionation?).

8. Minor Comment: I noticed that five different FAIMS voltages were used. Were they used within a single MS run, and if so, have you optimized the CV voltages? I would have assumed five is already too many, even for the Astral, as it takes too much time to switch. We usually tend to use three for TMT samples in our lab.

(Remarks on code availability)

Code availability: This reviewer has never used figshare before, so please accept my apologies if the following is my fault: When searching for the given dataset that should contain the code (10.6084/m9.figshare.28016501), more than 10k results are shown and at least the top 10 do not seem to correlate to this study. No figshare entry with that exact number can be found in this reviewers' hands, is it public already? Access to the raw data in PRIDE works perfectly fine.

Reviewer #2

(Remarks to the Author)

In this manuscript, Keele et al. investigated the proteome of three brain tissues (cortex, hippocampus, striatum) and kidney of 3, 12, and 20 months old mice. The authors used latest Orbitrap Astral mass spectrometer and demonstrated how it can improve the detection of age-related changes in protein expressions. Although I'm not very familiar with the technical aspects of mass spectrometry, I believe that the analysis pipeline demonstrated by the authors in this study could be a valuable reference for other researchers who want to perform TMT-based proteomic analysis with Orbitrap Astral. On the other hand, in regard to the biological findings, although the current study may have detected more age-dependent proteins in the brain and kidney than previous studies, I do not feel that the dataset itself represents a significant advancement for the field of aging research, due to the concerns described below.

1. Evaluation and demonstration of TMT-based proteomics using Orbitrap Astral are informative. However, I could not find C# program developed by the authors for this analysis. The authors should disclose this program.

2. Latest model of mass spectrometry (Orbitrap Astral) allowed the authors to better characterize the age-related changes in protein expressions up until middle age. However, this does not mean that protein expression changes during aging has been better characterized in this study than before. Changes associated with aging, or becoming geriatrics, can be adequately investigated only with inclusion of specimens collected from genuinely old organisms. Many studies investigating mouse aging proteome included 24 ~ 30 months old mice in their analysis. The authors cannot conclude that they identified more age-related changes than before simply by the number of statistically significant age-dependent proteins. This largely depends on the samples, methods and criteria for identifying such proteins.

3. A comparison of the number of proteins whose abundance was different between 8- and 18-months-old mice and the number of proteins whose abundance was different among 3-, 12-, and 20-months-old mice is meaningless. It is also totally expected that the number of age-dependent protein is higher in the later case. It is likely that this is mostly due to the usage of significantly younger mice (3- vs. 8-months-old mice). However, The finding that many previously undetected proteins showed age-related expression changes is interesting. In regard to this point, the authors should compare their data with data of various aging proteome studies and not only with the data obtained in their previous study, which only used non-geriatric mice.

4. Even if the current study detected more age-dependent proteins in brain and kidney, the study lacks new biological insight. Many similar datasets are already available. Especially, a paper previously published by the authors in Cell Reports two years ago (PMID: 37405913) provided a resource of aging proteome for 10 tissues, which is more comprehensive than the current study. In order to confer sufficient novelty to the current study, the authors should show how improved depth of proteomics contributed to the understanding of aging process in the brain and kidney with experimental validation.

5. The authors compared their aging proteome data with the early developmental proteome data reported by Wang et al. However, these two datasets are from different strains. It is well know that different strains (for example C57BL/6J and C57BL/6N) age in different ways. Age-related gene expression changes can significantly differ between different studies even in the same strain. The authors noted that "Fundamental differences remain between the studies (e.g., different institutions at different times, sister strains of mice, differences in tissue collection, and DIA versus TMT proteomics", but did not take this point into consideration in the following analysis. Without adjustment or at least convincing evaluation of the

effects of these factors, it would be premature to draw any conclusion in regard to the relationships between development and aging.

(Remarks on code availability)

Reviewer #3

(Remarks to the Author)

Despite several studies, proteomic coverage of various tissues, particularly at old age, remains relatively narrow, especially across both sexes. This study introduces an improved method for proteomic analysis by integrating Orbitrap Astral Mass Spectrometry with Tandem Mass Tag (TMT) technology, enhancing coverage and accuracy across multiple tissues in mice. The researchers analyzed three age groups (3, 12, and 20 months) and both sexes, covering brain tissues (cortex, hippocampus, striatum) and kidney, identifying ~9,000 proteins per tissue. To improve quantification accuracy, they developed a peptide filtering strategy to eliminate low-resolution and low signal-to-noise peptides. Their analysis identified both linear (continuous) and non-linear (non-continuous) age trajectories, revealing both sex and age differences in the kidney, while brain tissues showed primarily age-related changes with minimal sex differences. In addition, PCA indicated that tissue type was the strongest driver of protein variation (97.5%), followed by sex (0.4%) and age (0.5%). While some of these findings are important, particularly the potential to detect age-related changes earlier in life, the study's contribution to the aging field remains limited. Therefore, at its current stage, publication in a more specialized journal is recommended.

Point-by-Point Comments

1. Figures need to be chronological presented in the text for clarity.
2. Figure S3A: Show kidney results separately with distinct colors to highlight sex-based variations.
3. Figure S2: The use of Y chromosome-encoded genes as examples of sex-dependent differentially expressed genes adds minimal new knowledge.
4. Figure S4: The top enriched pathways in kidney involve cytoskeleton/actin, yet the discussion focuses on immune-related pathways, which are less novel. Explain this choice.
5. Comparison with Keele et al. (2023): Authors attribute their improved data to a broader age range (three groups vs. two). However, how do the MS corrections contribute beyond this? Clarify the added value of MS improvements vs. age group expansion.
6. Authors claim kidney clusters distinctly from brain tissues (Figure 5A), yet kidney also clusters more closely with striatum than with cortex/hippocampus (Figure 5B). Adjust the text and discuss this pattern.
7. Figure S3D: The comparison with Wang et al. (early development: 0.25, 1, and 2 months) is interesting but not obvious for aging analysis. Please expand this comparison and compare with additional proteomics data from these ages.
8. Discussion (Lines 503-504): Authors state that the "trend only becomes apparent with our broad age range", but three age groups do not constitute a truly broad range, for example please see SLAM project - <https://pubmed.ncbi.nlm.nih.gov/33211821/>. Please tone down this claim.
9. Abstract (Lines 17+): Mentions RNA splicing and transcription deterioration as aging factors, but these topics are not addressed in the paper. Consider removing or revising.

(Remarks on code availability)

Version 1:

Reviewer comments:

Reviewer #1

(Remarks to the Author)

The authors carefully considered all questions raised by the reviewers and resolved all major concerns. The quality of the manuscript is significantly improved, and this reviewer believes that especially the technological & methodological advancements presented will be of value for the proteomics community. The manuscript delivers only limited validation of novel age-related proteins nor interpretation of their biological relevance. However, the authors added a now clearly improved and more comprehensive comparison to findings from previous studies, which seems sufficient given the methodological focus of this study.

In conclusion I would recommend the revised manuscript for publication.

(Remarks on code availability)

Code found on figshare is in R and does not contain a readme with instructions for installing and running. How to run an R script is well documented in a multitude of other, external, resources though. It is well commented and easy to understand. The authors also precisely documented which script was used to generate the shown output for each figure in the manuscript. I did not test to re-run everything though.

Reviewer #2

(Remarks to the Author)

Unfortunately, I have to say that I'm disappointed with how the authors addressed the revisions for their manuscript. I clearly noted in the previous letter that "in regard to the biological findings, although the current study may have detected more age-dependent proteins in the brain and kidney than previous studies, I do not feel that the dataset itself represents a significant advancement for the field of aging research" and that "In order to confer sufficient novelty to the current study, the authors should show how improved depth of proteomics contributed to the understanding of aging process in the brain and kidney with experimental validation". However, the authors did not perform any experiment in their revision to extract new insights about aging from their proteomic data. Instead, they just compared their proteome data with additional public datasets to test consistency and differences among different datasets. This does not address my aforementioned major concern. I totally agree with Reviewer #3 that the study's contribution to the aging field is very limited and thus publication in a more specialized journal is recommended.

In addition, the combination of TMT-multiplexing and astral mass analyzer has been recently demonstrated in "Journal of Proteome Research" (PMID: 39937051). Statements regarding the methodological novelty also need to be soften.

(Remarks on code availability)

Reviewer #3

(Remarks to the Author)

The authors have addressed most of my comments. Regarding the detection of sex differences, it would be useful to analyze additional proteins beyond GSTA4. Additionally, the legends for Figure S3 are mixed up. Please correct this.

(Remarks on code availability)

Reviewer #1 (R1) summary: In their study, Keele and colleagues investigate brain development during aging, considering sex and brain-region differences in mice, thereby enhancing our understanding of aging processes. They compare their findings to a previous study focusing on infant mice. Notably, their work represents the first published study to perform isobaric multiplexing at the 18plex level using the new Orbitrap Astral MS, which will undoubtedly be of highest interest for the community from a technical and methodological perspective. As I am not a biologist, I cannot comment on the significance for research in brain development and aging, but the findings appear exciting to me.

While the authors discuss some existing limitations of their study, I have additional questions and comments that should be addressed prior to publication:

We express gratitude for Reviewer #1's interest and appreciation of our manuscript. In strong agreement with Reviewer #1, we believe this manuscript will be of value across a wide range of scientific fields for its novel contributions to mass spectrometry-based proteomics technology and methodology, as well as demonstrating statistical modeling of age changes in protein data and integration with other murine resource protein data sets to reveal global aging patterns. Reviewer #1's input has undoubtedly improved the quality of this manuscript.

R1 C1. Figure 1B: After counting, it appears that 18 signals are annotated in the spectrum, but not all correlate to the 18 expected reporter ions (RI). If I understand correctly, the authors describe detecting only 12 signals. I suggest aligning the text with the figure and annotating the visible RI signals, as some seem not fully resolved in the given example.

We appreciate the reviewer's suggestion. To improve clarity, we have revised Figure 1B and annotated the visible reporter ion signals. As described in the manuscript, the initial TMT reporter quantitation was performed using 0.003 Da peak match tolerance. Twelve of the eighteen ions in this scan have mass differences to its nearest TMT reporter ions smaller than 0.003 Da. In the following table, we annotated the Δ mass with their nearest TMT reporter ion. The other six are not considered as TMT reporter ions because their masses deviate by 0.003 Da from the expected reporter ion mass.

When we use 0.001 Da peak match tolerance for TMT reporter quantitation, only three ions were considered as reporter ions (highlighted in yellow, new **Fig. S1A**).

Reagent	Reporter Ion Mass	Detected	Δ mass
TMTpro-126	126.127726	126.1293	0.001574
TMTpro-127N	127.124761	127.1227	-0.002061
TMTpro-127C	127.131081	127.1311	0.000981

TMTpro-128N	128.128116	128.1254	-0.002716
TMTpro-128C	128.134436	128.1341	-0.000336
TMTpro-129N	129.131471		
TMTpro-129C	129.137791	129.1359	-0.001891
TMTpro-130N	130.134826		
TMTpro-130C	130.141146	130.1386	-0.002546
TMTpro-131N	131.138181	131.1411	0.002919
TMTpro-131C	131.144501		
TMTpro-132N	132.141536		
TMTpro-132C	132.147856	132.1459	-0.001956
TMTpro-133N	133.144891		
TMTpro-133C	133.151211	133.1491	-0.002111
TMTpro-134N	134.148246	134.1511	-0.002854
TMTpro-134C	134.154566		
TMTpro-135N	135.151601	135.1510	-0.000601

R1 C2. Resolution Issue: Likely caused by ion saturation, can the authors comment on the feasibility of using their TMT workflow for (ultra) low input samples, such as single cells or small laser microdissections of tissue where sample input is limited? I would assume fewer to no cases of ion saturation would be observed in such cases, while the extremely high sensitivity of the Astral MS would be highly beneficial (in addition to its speed) compared to Orbitrap-based TMTPro workflows. Have the authors tested lower inputs? If not, is it possible to test this (e.g., with sample leftovers that could be diluted)? If done, it would significantly enhance the manuscript's quality by opening a new use-case scenario for their workflow, even if it is just a proof-of-principle test showcasing the minimal input needed to obtain meaningful data.

Reviewer #1 makes an excellent point; their assumption is right. We have tested lower sample input amounts and analyzed the ion saturation issue. As the reviewer suggested, we desalted the remaining 12 fractions from the TMT experiment on the hippocampus and injected approximately half the amount used in the original run. This resulted in a reduced number of quantified peptides and fewer peptides with S/N greater than 1,440 across 18 channels (new **Fig. S1B**). The number of PSMs with any of the resolutions on TMT reporter ions below 45K decreased from 10,572 to 3,962, still impacting 1,383 proteins. Additionally, the number of quantified proteins dropped from 9,370 to 8,762 (new **Fig. S1C**).

Then, we selected one of the 12 fractions (A7) and injected various amount of material in a serial dilution manner, corresponding to approximately 1.0 μg , 0.5 μg , 0.25 μg , 0.125 μg , 0.0625 μg , and 0.03125 μg , respectively. Each raw file was analyzed and summarized to protein level individually. Peptide-spectrum matches (PSMs) with the total S/N lower than 1440 across 18 channels were filtered out. The results indicate that the number of PSMs with low resolution (any TMT reporter ion resolution below 45K) decreased dramatically with a lower sample input. The number of identified proteins decreased with decrease sample input (new **Fig. S1D-F**).

When A7 fraction (0.5 μg) was included in the analysis of all 12 fractions above, 153 PSMs from this fraction with lower resolutions were included in the protein quantitation, while only 10 PSMs were included in the single file analysis (**Fig. S1F**). This occurs because the linear discrimination analysis (LDA) was applied to individual files and across all twelve files, respectively.

These results indicate that reducing the injection amount will alleviate the ion saturation problem with less quantified proteins. Notably, PSMs with low resolutions still exist at lower injection amount, emphasizing the need to check resolution in experiments with reduced material input. For TMT experiments with limited injection material, such as single-cell proteomics, we also recommend checking the resolutions of the TMT reporter ions as the dynamic range of protein abundance is huge in cell or tissue samples.

We have added the results above in **Fig. S1** and the Results section (lines 282 to 289). In the discussion, we included the clarification:

“This issue is specific to the TMT and Orbitrap Astral combination and does not affect other DDA or DIA applications using other mass spectrometers. It can be mitigated by injecting less material; however, this approach may reduce the depth of protein quantitation.” (lines 549-551)

“Additionally, we recommend routinely performing Astral resolution checks and evaluating the resolution of TMT reporter ions acquired with the Orbitrap Astral Mass Spectrometer before proceeding with further data analysis to ensure optimal data accuracy and reliability. For TMT experiments with limited injection material, such as single-cell proteomics, we also recommend checking the resolutions of the TMT

reporter ions, given the substantial dynamic range of protein abundance in cellular and tissue samples.” (lines 565-570)

R1 C3. Figure 2A & 2B: If I understand correctly, the main message is that RI from low and high-intensity peptides have a resolution problem, and filtering is needed for accurate quantification. However, this was hard to understand from the figure, and the figure legend reads more like the results section rather than a figure description. For example, log S/N was plotted according to the legend, but neither the x nor y axis is annotated with log S/N, which seems confusing at first glance. I recommend making the nomenclature in the legend and text consistent. Also, high S/N is used in Fig2D, but SN>1440 is used in 2B – I recommend clearly indicating if the same thing or something different is meant by “high.”

We appreciate Reviewer #1’s suggestion. We have made the nomenclature in the legend and text consistent. In Fig 2D, “high” refers to S/N >1440. We have unified the annotation to make the message clearer.

R1 C4. Figure 2D: This figure would benefit from better annotation. For example, instead of “count,” use something like “# proteins quantified” as axis annotation, or add a header as done in 2C, to make it easier to understand the figure at first glance. This also applies to Figure 2D/line 285 and following: It was not obvious to this reviewer what was filtered (shown in light blue) if not S/N & R. It was also hard to understand the difference from 2C/proteins. I assume “impact” here means different regulation of proteins that are still quantified? Please clarify the figure's appearance here.

We appreciate Reviewer #1’s suggestion. We have added a header in Fig 2D. In Figure 2D, the authors used S/N and R for filtering. We have made it clear in the main text (lines 557 to 560) and the figure legend.

R1 C5. Sample Replicates: If I understand correctly, samples were collected in triplicates for each condition. If so, please indicate what numbers are represented by the bars in Figure 2. Are they averages from n=3? If so, I recommend adding error bars and indicating the number of replicates used in the figure legend.

We thank Reviewer #1 for this question. The numbers presented in Fig. 2 were from each TMT experiment. Peptide filtering was performed at the MS2 level and had no effect on the quantitation of the replicates, as TMT reporter ion intensities were acquired with each MS2 scan.

R1 C6. Figure S5: I appreciated the comparison to the previous study by the same first author, recorded on a Lumos. Can the authors include a comment on the overall protein numbers and overlap found in their previous study compared to this one? How significant is the advantage they see on the Astral over the Lumos, given the need for filtering on the Astral? This information would be valuable for the community in deciding which instrument type might be best suited for their TMT project.

Reviewer #1 asks a great question about whether we can make recommendations for TMT studies for the Astral vs. Lumos based on the overall protein numbers observed. We refrained from over-emphasizing specific numbers in the initial manuscript due to concerns like those expressed by Reviewer #2 in that multiple factors differentiate this study from Keele *et al.* 2023, including mouse sub-strain, age groups, TMTplex, beyond just the mass spectrometer technology and the fact that these are simply different experiments run at different times. Furthermore, proteins were searched and summarized based on ENSEMBL protein ID in Keele *et al.* 2023 and Uniprot in the present study, which do not always have a

clean one-to-one mapping. A different filtering step in Keele et al. 2023 was also used: that study used TMT11plex and thus had two batches per tissue, which proteins were then filtered to those observed across both batches.

To get a better idea of how numbers compare, we performed a *post hoc* mapping of Uniprot IDs to ENSEMBL protein IDs, conservatively filtering to a single protein ID when a Uniprot ID mapped to multiple ENSEMBL protein IDs.

[Editorial note: this figure was redacted due to third-party rights. It can be found in Keele et al. 2023 Figure S1b].

Here we see that even after *post hoc* accounting for differences in how the proteins were summarized, the present study does appear to identify significantly more proteins per tissue (1,000 or more). In general, Orbitrap Astral Mass Spectrometer detects more proteins, particularly proteins of lower abundance (Fig. S6).

R1 C7. Workflow Comparison: In line with the value of instrument comparison discussed above, can the authors comment on the advantages and limitations they see within their TMT on Astral versus an LFQ-DIA on Astral approach, as performed by the cited Wang et al. study? The authors mention in their discussion: “This approach enabled us to overcome previous limitations in proteome coverage and depth and significantly boost the quantified protein number.” However, it is not fully clear to the reader which competing strategy this boost was compared to. Please accept my apologies if I missed this in the text. A direct comparison of TMT vs. DIA would be helpful for the reader to make an informed choice before starting a new study. This includes estimated coverage or required measurement time (I think the authors generated 96 fractions from their TMT mix, pooled to 24 fractions, of which 12 were measured; how would the required MS time compare to direct measurements in DIA mode without fractionation?).

We appreciate Reviewer #1’s suggestion. In this manuscript, we compared datasets acquired using the Orbitrap Lumos (Keele et al. 2023 and now Takasugi et al. 2024 as well) and demonstrated an improvement in proteome coverage, particularly for low-abundance proteins (Fig. S6). Despite the differences on experimental design, we believe the higher sensitivity and speed enabled higher coverage of the proteome.

TMT on Astral provides significant advantages for studies focusing on relative quantification across multiple conditions due to its multiplexing capabilities, allowing simultaneous analysis of several samples in one MS run, which increases throughput and efficiency. However, TMT can suffer from ratio compression and reporter ion interference, potentially impacting quantification accuracy, and the cost of tags along with extra sample preparation steps can be prohibitive. On the other hand, DIA on Astral provides comprehensive proteome coverage and sensitivity, offering a full scan of all detectable peptides in a sample, which might reduce the need for extensive fractionation and coincidentally decrease MS time. DIA's data acquisition method could mean fewer, or no fractions needed, simplifying the process but at the cost of increased computational complexity for data interpretation. In comparing TMT with DIA, our study notes an increase in proteome coverage and depth with TMT, but DIA could potentially match or exceed this without the need for extensive fractionation, though it lacks TMT's multiplexing advantages. Ultimately, the choice between these strategies should be guided by the specific scientific questions, available resources, and the balance between throughput, cost, and complexity of data analysis to ensure the method aligns with the experimental goals.

The direct comparison of TMT vs. DIA is beyond the scope of this manuscript. We are aware of other studies performing this comparison, which are expected to be published online soon.

R1 C8. Minor Comment: I noticed that five different FAIMS voltages were used. Were they used within a single MS run, and if so, have you optimized the CV voltages? I would have assumed five is already too many, even for the Astral, as it takes too much time to switch. We usually tend to use three for TMT samples in our lab.(Remarks on code availability)

Thanks for Reviewer #1's comments. We have tested 5 CVs and 3 CVs. We found that it really depends on the setting of the experiment, e.g., fractionated sample vs non-fractionated sample, sample abundance, column, and HPLC gradient. Based on our analysis, using five CVs performed as well as, if not better than, three CVs in our TMT workflow with fractionated samples.

R1 C9. Code availability: This reviewer has never used figshare before, so please accept my apologies if the following is my fault: When searching for the given dataset that should contain the code (10.6084/m9.figshare.28016501), more than 10k results are shown and at least the top 10 do not seem to correlate to this study. No figshare entry with that exact number can be found in this reviewers' hands, is it public already?

Access to the raw data in PRIDE works perfectly fine.

We apologize. The full DOI URL is <https://doi.org/10.6084/m9.figshare.28016501>. We have now included it in the manuscript.

Reviewer #2 (R2) summary: In this manuscript, Keele et al. investigated the proteome of three brain tissues (cortex, hippocampus, striatum) and kidney of 3, 12, and 20 months old mice. The authors used latest Orbitrap Astral mass spectrometer and demonstrated how it can improve the detection of age-related changes in protein expressions. Although I'm not very familiar with the technical aspects of mass spectrometry, I believe that the analysis pipeline demonstrated by the authors in this study could be a valuable reference for other researchers who want to perform TMT-based proteomic analysis with Orbitrap Astral. On the other hand, in regard to the biological findings, although the current study may have detected more age-dependent proteins in the brain and kidney than previous studies, I do not feel that the dataset itself represents a significant advancement for the field of aging research, due to the concerns described below.

We thank Reviewer #2 for their time and interest in our work. They agree with Reviewer #1 in terms of the value of this work for mass spectrometry-based proteomics as a demonstration of how to use the Astral Orbitrap technology for large-scale quantitative protein studies. They raise reasonable concerns about the limitations about the downstream analysis of aging changes, comparisons to other studies, and inherent limitations in interpreting differences between studies. We believe these limitations are largely addressable through proper communication of the requisite assumptions, which should greatly improve this manuscript and ensure that readers do not over interpret its results. We have also added a comparison to an additional, highly relevant study, Takasugi et al. 2024. Notably, this study also included geriatric mice (30 months-old). We saw even better consistency than with Keele et al. 2023 and allowed us to identify proteins with age trends that extended into geriatric ages. Finally, integrating these diverse data resources has enabled us to uncover developmental and aging trends that were previously inaccessible through high-throughput quantitative proteomics studies.

R2 C1. Evaluation and demonstration of TMT-based proteomics using Orbitrap Astral are informative. However, I could not find C# program developed by the authors for this analysis. The authors should disclose this program.

We have now made the C# program available in the figshare repository (<https://doi.org/10.6084/m9.figshare.28016501>).

R2 C2. Latest model of mass spectrometry (Orbitrap Astral) allowed the authors to better characterize the age-related changes in protein expressions up until middle age. However, this does not mean that protein expression changes during aging have been better characterized in this study than before. Changes associated with aging, or becoming geriatrics, can be adequately investigated only with inclusion of specimens collected from genuinely old organisms. Many studies investigating mouse aging proteome included 24 ~ 30 months old mice in their analysis. The authors cannot conclude that they identified more age-related changes than before simply by the number of statistically significant age-dependent proteins. This largely depends on the samples, methods and criteria for identifying such proteins.

We appreciate Reviewer #2's concern that our claim regarding better characterization of protein expression changes with aging may be too strong. We agree that the ability to characterize aging-related proteomic changes depends on multiple factors, including sample selection, the range of age groups, organismal differences, and the analytical methods used.

We aimed to highlight the specific improvements in our study through comparing to Keele et al. 2023. We were particularly motivated by the striking difference in the number of age-related proteins detected, especially in the brain, in these two studies. In line with Reviewer 2's comment, the proteins observed in only this study had lower average S/N and a lower rate of age differences detected compared to proteins observed in both studies (**Fig. S6A-C**). These findings suggest that the identification of more proteins with age differences in our study is likely due, in part, to improved statistical power due to observing a greater dynamic range of ages (three age groups rather than two), range rather than solely to advances in mass spectrometry technology. Meanwhile, the inclusion of three distinct age groups in our study provided the resolution to capture non-linear age dynamics that would be impossible to detect with only two age groups. These results highlighted the importance of including more age groups to better characterize age-related changes.

To address Reviewer #2 comments regarding investigating mouse aging proteome by including 24 ~ 30 months old mice, we have expanded our analyses to integrate data from Takasugi et al. 2025, which includes geriatric male mice. This addition has allowed us to extend our findings and highlight proteins with notable aging trends that persist in geriatric stages (lines 397- 411, **Fig 3G-H, Fig. S5D-F**)

R2 C3. A comparison of the number of proteins whose abundance was different between 8- and 18-months-old mice and the number of proteins whose abundance was different among 3-, 12-, and 20-months-old mice is meaningless. It is also totally expected that the number of age-dependent protein is higher in the later case. It is likely that this is mostly due to the usage of significantly younger mice (3- vs. 8-months-old mice). However, The finding that many previously undetected proteins showed age-related expression changes is interesting. In regard to this point, the authors should compare their data with data of various aging proteome studies and not only with the data obtained in their previous study, which only used non-geriatric mice.

We appreciate Reviewer #2’s perspective and agree that a direct comparison between the present study and Keele et al. (2023) comes with important caveats that should be clearly communicated. We believe this comparison remains valuable, as the previous study reported relatively few age-related differences in brain tissues. Our motivation for making this comparison was to highlight how the inclusion of additional age groups, particularly younger adult mice (3-month-old), contributed to the differences observed. In this regard, we fully agree with Reviewer #2’s conclusion that the expanded age range plays a key role in detecting more age-dependent proteins. Importantly, we also note that many of the age-related changes observed in our study do not appear to be restricted to younger adulthood. For example, as shown in **Fig. 4C**:

If most of the new changes stemmed entirely from 3M compared to 12M and 20M, we would expect most age trends to be categorized as Up/Down-Flat. Generally, those do appear to be more frequent than the Flat-Up/Down trends, but not more so than the linear trends of Up-Up and Down-Down. In terms of overall numbers, the young age group is allowing us to more confidently detect linear trends that continue into later adulthood rather than specifically changes between young adulthood (3M) and middle adulthood (12M). We believe this more nuanced modeling of age changes (continuous vs non-continuous) is a novel feature of this work.

Similarly demonstrating that we are overall in alignment with Reviewer #2’s view on not focusing on counts of statistically significant differences, in **Fig 4C-F**, we highlight that there are consistent

changes between the studies, in terms of sex, age, and age-by-sex. This is meant as validation of age changes across studies rather than assessing one study/analysis as being superior to the other.

Reviewer #2 suggests comparing our results to additional studies that include geriatric mice. To address this, we have conducted a detailed comparison with Takasugi et al. (2024), a recent study that includes geriatric mice (30 months old), though only males. Despite differences in experimental design, such as sub-strain variations and tissue selection, we observe strong validation across these studies (**Fig. 3G-H, Fig. S7D-F**). Moreover, this additional dataset further supports our conclusion that aging-related proteomic trends in adulthood and geriatric stages tend to follow more linear or continuous trajectories compared to early developmental changes (**Fig. 5A-B**). Importantly, it also allows us to confirm that select protein expression trends extend into geriatric life, reinforcing the biological relevance of our findings (**Fig. 6J**).

R2 C4. Even if the current study detected more age-dependent proteins in brain and kidney, the study lacks new biological insight. Many similar datasets are already available. Especially, a paper previously published by the authors in Cell Reports two years ago (PMID: 37405913) provided a resource of aging proteome for 10 tissues, which is more comprehensive than the current study. In order to confer sufficient novelty to the current study, the authors should show how improved depth of proteomics contributed to the understanding of aging process in the brain and kidney with experimental validation.

We acknowledge Reviewer #2's point that this work is not as comprehensive of a resource as Keele *et al.* 2023, Wang *et al.* 2024, and Takasugi *et al.* 2024, in terms of the number of tissues profiled. And we would like to emphasize that our contributions extend beyond generating aging proteomics datasets. Specifically, we (1) provide the proteomics community with a standardized data analysis workflow for TMT datasets generated on the Orbitrap Astral MS, ensuring more accurate and comprehensive proteomic profiling, and (2) introduce novel approaches for modeling age-related changes (e.g., continuous vs. non-continuous trajectories) and integrating findings across studies spanning different lifespan periods.

Regarding Reviewer #2's point on experimental validation of newly identified signals, we have validated age-related changes through comparisons with multiple prior studies (Keele *et al.*, 2023; Tsumagari *et al.*, 2023; and Takasugi *et al.*, 2024). These comparisons allow us to detect age-related patterns that are consistent with our findings, even if they did not reach statistical significance in previous studies.

While experimental validation of novel age-related proteins is an important next step, it falls outside the scope of this study. We anticipate that our findings will serve as a valuable resource for hypothesis generation, supporting future investigations in aging research laboratories.

R2 C5. The authors compared their aging proteome data with the early developmental proteome data reported by Wang *et al.* However, these two datasets are from different strains. It is well known that different strains (for example C57BL/6J and C57BL/6N) age in different ways. Age-related gene expression changes can significantly differ between different studies even in the same strain. The authors noted that "Fundamental differences remain between the studies (e.g., different institutions at different times, sister strains of mice, differences in tissue collection, and DIA versus TMT proteomics)", but did not take this point into consideration in the following analysis. Without adjustment or at least convincing evaluation of the effects of these factors, it would be premature to draw any conclusion in regard to the relationships between development and aging.

We appreciate the reviewer for raising the issue of C57BL/6 sub-strains, as it led us to recognize that we had not explicitly noted that comparisons between our study and those by Keele *et al.* (2023) and Takasugi *et al.* (2024) also involve different C57BL/6 sub-strains (C57BL/6J vs. C57BL/6N). To address

this, we now cite studies highlighting differences between these sub-strains (Simon et al., 2013), including their potential impact on aging (Yamada et al., 2025). We acknowledge the importance of specifying genetic background across studies to provide proper context for comparisons.

Due to the inherent confounding of sub-strain differences (along with other experimental variables) within individual studies, we did not perform joint modeling of datasets across different sub-strains. Instead, we employed a meta-analysis framework, where studies are aligned based only on summary statistics (regression coefficients). Importantly, we observe strong validation of age- and sex-related differences across studies of different C57BL/6 sub-strains in adulthood, suggesting that while early developmental differences across sub-strains cannot be fully ruled out, it is unlikely that adult aging processes would be highly consistent while early developmental trajectories would diverge significantly.

Our conclusions are intentionally framed at a broad, systems-level perspective. For instance, the significantly greater extent of non-continuous age changes—primarily driven by early postnatal development—compared to adulthood is evident in both overall variation (as seen in principal components) and individual protein-level changes. It seems biologically plausible that major proteomic shifts occur during early development, whereas age-related changes in adulthood involve different sets of proteins and more gradual transitions. Additionally, sex differences in kidney proteins were highly consistent across studies and became more pronounced after puberty. Furthermore, synapse-related dynamics (increased in early life, decreased in adulthood) were replicated in an external synaptome study and partially validated in another proteomics study (Tsumagari et al., 2023). We now provide specific examples of proteins illustrating these trends across studies (Fig. 6E, J).

We fully acknowledge the reviewer's point and have made revisions to emphasize that we cannot entirely rule out sub-strain contributions to observed differences. However, given the broad and biologically consistent patterns we detect—many of which align with well-established developmental and aging processes—it seems highly unlikely that these system-wide trends are primarily driven by sub-strain identity, unless the sub-strains exhibited dramatically different phenotypes beyond the behavioral differences reported in the literature.

Reviewer #3 (R3) summary: Despite several studies, proteomic coverage of various tissues, particularly at old age, remains relatively narrow, especially across both sexes. This study introduces an improved method for proteomic analysis by integrating Orbitrap Astral Mass Spectrometry with Tandem Mass Tag (TMT) technology, enhancing coverage and accuracy across multiple tissues in mice. The researchers analyzed three age groups (3, 12, and 20 months) and both sexes, covering brain tissues (cortex, hippocampus, striatum) and kidney, identifying ~9,000 proteins per tissue. To improve quantification accuracy, they developed a peptide filtering strategy to eliminate low-resolution and low signal-to-noise peptides. Their analysis identified both linear (continuous) and non-linear (non-continuous) age trajectories, revealing both sex and age differences in the kidney, while brain tissues showed primarily age-related changes with minimal sex differences. In addition, PCA indicated that tissue type was the strongest driver of protein variation (97.5%), followed by sex (0.4%) and age (0.5%). While some of these findings are important, particularly the potential to detect age-related changes earlier in life, the study's contribution to the aging field remains limited. Therefore, at its current stage, publication in a more specialized journal is recommended.

We appreciate Reviewer #3's accurate summary of the manuscript. They clearly dedicated time and energy evaluating our work, which will ultimately significantly improve its final form. We respectfully disagree with the view that this work does not make enough of a contribution to warrant publication in *Nature Communications*. In part, our revisions, influenced by Reviewer #3 and detailed below, improve the quality and impact of the work. But even before the revisions, we hope Reviewer #3 will consider that this work appeals to a broad scientific audience, making contributions to the proteomics field through

advancement in mass spectrometry data analysis workflow. It also provides resource data for future aging studies, particularly in the context of the brain, and provides novel insights into modeling age changes and integration across studies. This analysis revealed insights into aging/developmental patterns, that to our knowledge, has not been observed previously based on quantitative proteomics data. We hope to better communicate this last point in the revised manuscript.

R3 C1. Figures need to be chronological presented in the text for clarity.

We appreciate Reviewer #3's suggestion and understand that they may be referring to a strictly sequential order (e.g., 1–7, S1–S8). Our initial intention was to order the figures based on their first reference in the text. However, we recognize the importance of maintaining a clear and consistent structure, and we sincerely apologize for any confusion. In response to this feedback, we have now revised the figure numbering to follow a sequential order throughout the manuscript.

R3 C2. Figure S3A: Show kidney results separately with distinct colors to highlight sex-based variations.

We thank Reviewer #3 for this suggestion. Please see the updated **Fig. S4A**, which we believe strongly improves the clarity of the effect of sex on overall kidney protein variation.

R3 C3. Figure S2: The use of Y chromosome-encoded genes as examples of sex-dependent differentially expressed genes adds minimal new knowledge.

We agree with Reviewer #3 that detecting a sex difference for a gene encoded on the Y chromosome is not particularly novel. The other example protein, GSTA4 (new **Fig. S3E**), is an autosomal gene but also known to have elevated expression in females, which we now cite. We have added extra detail in the Results section (lines 346-349). We aimed to illustrate the data quality by showing these examples, given their value as a resource for future research.

R3 C4. Figure S4: The top enriched pathways in kidney involve cytoskeleton/actin, yet the discussion focuses on immune-related pathways, which are less novel. Explain this choice.

We appreciate the reviewer's insights and acknowledge the previously reported aging-related changes in the actin cytoskeleton, such as those described by Takemon et al. 2021, which we have already cited in the manuscript. As with our response to the previous comment, our supplementary figures were primarily intended to highlight known features that demonstrate data quality and consistency across proteomic datasets. We originally selected the immune gene set for its broader relevance and appeal to a general scientific audience. However, upon revisiting this Results section in light of Reviewer #3's comments, we recognized an opportunity to enhance it with additional details and examples.

To address this, we have updated Figure S5 to include a volcano plot for the actin cytoskeleton findings (now **Fig. S5B**), while the innate immune response is now presented in Fig. S5C. We have also revised the corresponding text in the Results section (lines 352–355) and provided additional details on USP24, the example gene with an age-by-sex difference in the cortex (**Fig. S3F**). We sincerely appreciate Reviewer #3's feedback, which has significantly improved this section, particularly in reinforcing its value as a resource for future research.

R3 C5. Comparison with Keele et al. (2023): Authors attribute their improved data to a broader age range (three groups vs. two). However, how do the MS corrections contribute beyond this? Clarify the added value of MS improvements vs. age group expansion.

We appreciate Reviewer #3's request for clarification regarding the relative contributions of MS improvements versus age group expansion.

In this study, we developed a filtering strategy in this study to remove saturated ions that could interfere with protein quantitation in TMT datasets acquired using the Orbitrap Astral MS, providing a standard workflow to the proteomics field. Based on the data analysis, we revealed that the Orbitrap Astral MS provides a significant advantage in detecting low-abundance proteins (see our response to R1 C6, **Fig. S6**), allowing for a deeper and more comprehensive survey of the proteome. We found that most proteins detected with age differences in our study were quantified in the previous study with two age groups. Furthermore, the proteins observed in only this study had lower average S/N and a lower rate of age differences detected compared to proteins observed in both studies (**Fig. S6A-C**). Here, we would like to highlight that the low-abundance proteins uniquely detected in this study also exhibit age-related changes, expanding the coverage of age-related proteins. Together, the combination of advanced MS technology and a more refined experimental design significantly improved our ability to capture age-related proteomic changes in this study. We recognize that our original statement may have been unclear, and we have revised it for clarity in lines 381–384.

R3 C6. Authors claim kidney clusters distinctly from brain tissues (Figure 5A), yet kidney also clusters more closely with striatum than with cortex/hippocampus (Figure 5B). Adjust the text and discuss this pattern.

We appreciate Reviewer #3's careful examination of Figure 5A-B and understand the concern raised. However, we believe there may be a slight misinterpretation of these figures. Based on the hierarchical clustering and resulting dendrogram, kidney represents the overall basal lineage for both age- and sex-related differences. In the clustering, the kidney is equally separated from all three brain tissues. The relative positioning of the more basal brain tissue lineage (hippocampus for sex differences and striatum for age differences) in relation to kidney is likely arbitrary due to the hierarchical clustering approach. For this reason, we have not adjusted the Results but would be happy to clarify further if needed.

R3 C7. Figure S3D: The comparison with Wang et al. (early development: 0.25, 1, and 2 months) is interesting but not obvious for aging analysis. Please expand this comparison and compare with additional proteomics data from these ages.

We appreciate Reviewer #3's interest in our integration of early developmental proteomics data from Wang et al., which we believe represents a novel and valuable aspect of this manuscript. This approach allows us to capture a more comprehensive view of protein dynamics across the lifespan in both the brain and kidney. While it is not as powerful as a full quantitative proteomics experiment spanning six or more age groups, it provides important insights into lifespan-wide protein changes.

In response to the reviewer's suggestion, we have expanded this section of the Results to include illustrative examples (**Fig. 6E, J**) that further clarify our observations. Additionally, we now incorporate a comparison with a newly available adult mouse proteomics dataset from Takasugi et al. (2024), which is particularly exciting as it includes a geriatric age group (30 months old). Notably, this dataset strongly aligns with our findings (**Fig. 3G-H**) and supports our observation that age-related proteomic changes in adulthood follow a more linear and continuous trajectory (**Fig. 4B**), in contrast to the more distinct shifts observed in early development (**Fig. 6B**). Furthermore, we confirm that key example proteins exhibit age-related changes that extend into geriatric stages, further validating our approach.

R3 C8. Discussion (Lines 503-504): Authors state that the "trend only becomes apparent with our broad

age range", but three age groups do not constitute a truly broad range, for example please see SLAM project - <https://pubmed.ncbi.nlm.nih.gov/33211821/>. Please tone down this claim.

We appreciate Reviewer #3's feedback and agree that our original statement was too strong. We have revised it to 'broader age range' to more accurately reflect our comparison with Keele et al. (2023), which included only two age groups. We fully acknowledge the need for more comprehensive aging resources in quantitative proteomics, with a finer age group resolution like that of the SLAM project.

R3 C9. Abstract (Lines 17+): Mentions RNA splicing and transcription deterioration as aging factors, but these topics are not addressed in the paper. Consider removing or revising.

We agree that these topics were not directly addressed in the manuscript. To ensure alignment between the abstract and the main text, we have removed this mention. Thank you for your thoughtful feedback.

Reviewer #1 remarks to the authors: The authors carefully considered all questions raised by the reviewers and resolved all major concerns. The quality of the manuscript is significantly improved, and this reviewer believes that especially the technological & methodological advancements presented will be of value for the proteomics community. The manuscript delivers only limited validation of novel age-related proteins nor interpretation of their biological relevance. However, the authors added a now clearly improved and more comprehensive comparison to findings from previous studies, which seems sufficient given the methodological focus of this study. In conclusion I would recommend the revised manuscript for publication.

We appreciate Reviewer #1's assessment of the value of this work to the proteomics field. They also note that we have improved the manuscript through more comprehensive comparison to previous studies, which we note does have highlight system-wide shifts in protein levels across the mouse lifespan. We are grateful for their feedback throughout the review, which has greatly increased the quality of this manuscript.

Reviewer #1 remarks on code availability: Code found on figshare is in R and does not contain a readme with instructions for installing and running. How to run an R script is well documented in a multitude of other, external, resources though. It is well commented and easy to understand. The authors also precisely documented which script was used to generate the shown output for each figure in the manuscript. I did not test to re-run everything though.

We are glad that Reviewer #1 appreciates the setup up the figshare repository. We have also now added a README text file to describe the repository and provide instructions for re-running analyses and generating published results and figures.

Reviewer #2 remarks to the authors: Unfortunately, I have to say that I'm disappointed with how the authors addressed the revisions for their manuscript. I clearly noted in the previous letter that "in regard to the biological findings, although the current study may have detected more age-dependent proteins in the brain and kidney than previous studies, I do not feel that the dataset itself represents a significant advancement for the field of aging research" and that "In order to confer sufficient novelty to the current study, the authors should show how improved depth of proteomics contributed to the understanding of aging process in the brain and kidney with experimental validation". However, the authors did not perform any experiment in their revision to extract new insights about aging from their proteomic data. Instead, they just compared their proteome data with additional public datasets to test consistency and differences among different datasets. This does not address my aforementioned major concern. I totally agree with Reviewer #3 that the study's contribution to the aging field is very limited and thus publication in a more specialized journal is recommended.

In addition, the combination of TMT-multiplexing and astral mass analyzer has been recently demonstrated in "Journal of Proteome Research" (PMID: 39937051). Statements regarding the methodological novelty also need to be soften.

We are sincerely grateful for Reviewer #2's perspective and thoughtful feedback. As stated in our initial revision, we fully agree that experimental validation is critical for advancing aging research, especially when moving from large-scale, hypothesis-generating studies to fine-grained mechanistic insights. However, we believe that it is beyond the realistic scope of a single manuscript to comprehensively address advancements in proteomics methodology, introduce novel statistical modeling and integration with prior datasets, and also include detailed experimental follow-up to dissect specific aging mechanisms.

We would also like to note that Reviewer #3 encouraged further comparisons with previous studies, which we have incorporated to strengthen the validation of age-associated proteomic changes.

Additionally, we have included references to recently published papers utilizing the Orbitrap Astral mass spectrometer and have adjusted the language in our manuscript accordingly. These studies, likely under review concurrently with ours, highlight the relevance of this area of research to the field of proteomics.

Once again, we deeply appreciate the constructive input, which has challenged us to critically re-express and refine our work. We believe these revisions have significantly improved the manuscript.

Reviewer #3 remarks to the authors: The authors have addressed most of my comments. Regarding the detection of sex differences, it would be useful to analyze additional proteins beyond GSTA4. Additionally, the legends for Figure S3 are mixed up. Please correct this.

We thank Reviewer #3 for their constructive feedback, which has helped improve the quality of our manuscript. In response, we have made the complete set of results related to sex differences publicly available, including analyses of proteins beyond GSTA4 and functional/pathway-level insights derived from GSEA. Additionally, we have corrected the Figure S3 legends as requested.